# An Analysis of Elo Rating Systems via Markov Chains

**Sam Olesker-Taylor**
Department of Statistics
University of Warwick
Coventry, CV4 7AL, UK
sam.olesker-taylor@warwick.ac.uk

**Luca Zanetti**
Department of Mathematical Sciences
University of Bath
Bath, BA2 7AY, UK
lz2040@bath.ac.uk

## Abstract

We present a theoretical analysis of the Elo rating system, a popular method for ranking skills of players in an online setting. In particular, we study Elo under the Bradley–Terry–Luce model and, using techniques from Markov chain theory, show that Elo learns the model parameters at a rate competitive with the state of the art. We apply our results to the problem of efficient tournament design and discuss a connection with the fastest-mixing Markov chain problem.

## 1 Introduction

The Elo rating system is a popular method for calculating the relative skills of players (or teams) in sports analytics and particularly chess [2, 4, 1, 3]. It is based on a simple zero-sum update rule: if player $i$ beats player $j$, then the rating of player $i$ increases proportionally to the model probability that $i$ would lose to $j$, while the rating of $j$ decreases by the same amount. This amount depends on the previously estimated difference in skills between $i$ and $j$.

Despite their widespread popularity, Elo rating systems still lack a rigorous theoretical understanding [6]. Here, we take a probabilistic approach and study Elo under the well-known Bradley–Terry–Luce model (BTL) [11, 28]. In this model, the probability $p_{i,j}$ that $i$ wins against $j$ is $w_i/(w_i + w_j)$, where $w_k$ is the *strength* of player $k$. In Elo, this is usually reparametrised via $w_k = \mathrm{e}^{\rho_k}$:

$$p_{i,j} = \mathrm{e}^{\rho_i}/(\mathrm{e}^{\rho_i} + \mathrm{e}^{\rho_j}) = 1/(1 + \mathrm{e}^{\rho_j - \rho_i}) = \sigma(\rho_i - \rho_j),$$

where $\rho_k$ is the *true rating* of $k$ and $\sigma(z) := 1/(1 + \exp(-z))$ for $z \in \mathbb{R}$ is the sigmoid function. In this setting, after observing $i$ beat $j$, the corresponding Elo ratings $x_i$ and $x_j$ are updated as

$$x_i \leftarrow x_i + \eta\sigma(x_j - x_i) \quad \text{and} \quad x_j \leftarrow x_j - \eta\sigma(x_j - x_i),$$

where the step-size $\eta > 0$ is chosen by the modeller. The size of the update depends exponentially on the difference in ratings: beating a much lower rated opponent does not change the ratings much.

The goal of the Elo rating system is to estimate the true ratings of $n$ players by observing results of matches between pairs of players. It is, therefore, aiming to solve the problem of ranking from pairwise comparisons. Compared with most algorithms in the area [5, 20, 27, 31, 37], however, Elo benefits from three qualities that help explain its popularity in real-world applications: simplicity, interpretability and the ability to update a ranking of the players in an *online* fashion.

The knowledgeable reader might have noticed that Elo's update rule is actually based on the gradient of the BTL log-likelihood: Elo can be interpreted simply as stochastic gradient descent with fixed step-size. Rather than studying Elo from a convex optimisation angle, however, we take a more probabilistic point of view: we assume at each time $t$, players $i$ and $j$ are selected to play against one another with probability $q_{i,j}$. This allows us to interpret Elo as a Markov chain over $\mathbb{R}^n$ and deploy powerful tools to study its behaviour. On the other hand, this approach also presents us with some challenges: Elo is not a reversible Markov chain and, while it has a unique stationary distribution, assuming a minor and natural condition on $(q_{i,j})_{i,j}$ [7], it does not converge to it in total variation [6].

## 1.1 Our Results

Our main contribution (Theorem 2.5) shows that Elo ratings, averaged over time, well-approximate the true ratings of the players with high probability. In order to avoid the potential of unbounded ratings, which are unrealistic in practice, we consider a variant of Elo in which ratings are capped, whilst maintaining their zero-sum property. We obtain rates of convergence with respect to the number of observed matches that are competitive against the state of the art in the BTL literature. In contrast to most other algorithms for the BTL model studied in the literature, Elo learns the parameters of a BTL model in an online fashion. These rates of convergence is remarkable since Elo was originally conceived as a simple ranking system for chess players. Furthermore, our approach is very robust, and also applies to a *parallel* set-up in which multiple games are played concurrently.

We also discuss the problem of *tournament design*: we assume we are given a tournament (comparison) graph $G = ([n], E)$, and we would like to choose the match-up probabilities $(q_{i,j})_{\{i,j\} \in E}$ so that Elo's convergence rate is maximised. We highlight a connection between this problem and that of finding the *fastest mixing Markov chain on $G$* [10, 39], where the corresponding Markov chain on the graph is either in discrete or continuous time depending on whether we optimise number of *games* or *parallel rounds*, respectively. As far as we know, this connection has not been made formally before in the sequential set-up, and both the analysis and optimisation of the parallel set-up are new. In §4, we also provide experimental results that showcase the usefulness of our strategy.

## 1.2 Related Work

Despite Elo ratings being the standard ranking system in many sports analytics communities [38], there is a scarcity of work analysing Elo from a probabilistic perspective. In particular, we are aware only of [6], and the unpublished notes [7] by the same author, in which Elo is studied as a Markov process and convergence in distribution to a unique stationary distribution is proved.

If we consider Elo as a technique to estimate the parameters of the BTL model, there is a wealth of recent literature on the topic by the machine learning community [20, 37, 31, 5, 27, 9]. In contrast to our setting, however, previous work typically considers an *offline* scenario, in which the ratings of the players are computed after all the scheduled matches have taken place. By standard concentration inequalities, this allows one to obtain a very good approximation of the probability a player wins against their neighbours in the comparison graph. The goal is then to deduce a global ranking of the players from such (very good) local information. In our setting, Elo ratings are dynamically updated before a good local approximation is achieved, which makes the analysis more challenging and requires the use of powerful, but delicate, concentration inequalities for Markov chains.

A comparison between the rate of convergence for Elo ratings obtained in our work and the rate of convergence of other algorithms for the BTL model is discussed in §2.

Finally, we mention that the connection between Markov chains and stochastic gradient descent with fixed step size has been studied before, e.g., in [17].

## 2 Convergence Rates of Elo Ratings

In this section, we state our main result on the convergence rate of the time-averaged Elo ratings, discuss related work and outline the most important parts of the proof.

Throughout the paper, "$f \lesssim g$" means "$f = O(g)$", "$f \ll g$" means "$f = o(g)$" and "whp" means "with probability $1 - O(1/n)$"; the notation $\tilde{O}(\cdot)$ hides logarithmic factors; finally, $\mathbb{E}_\pi[\cdot]$ indicates that $X^0 \sim \pi$.

## 2.1 Our Results

We start with the explicit definition of our Markov chain.

**Definition 2.1** (Elo Process). Let $M \in \mathbb{R}$ and $n \geq 2$. Let $\rho \in [-M, M]^n$ with $\sum_k \rho_k = 0$. Let $q$ be a distribution on unordered pairs in $[n]$. Let $\eta \in (0, \frac{1}{4})$. A step of $\text{Elo}_M(q, \rho; \eta)$ proceeds as follows.

    0. Suppose that the current vector of ratings is $x \in \mathbb{R}^n$.

1. Choose unordered pair $\{I, J\}$ to play according to $q$:

$$\mathbb{P}[\{I, J\} = \{i, j\}] = q_{\{i,j\}} \quad \text{for all} \quad i, j \in [n].$$

2. Suppose that Player $I$ beats $J$, which has probability $\sigma(\rho_I - \rho_J)$. Update ratings $x_I$ and $x_J$:

$$x_I \leftarrow x_I + \eta\sigma(x_J - x_I); \quad x_J \leftarrow x_J - \eta\sigma(x_J - x_I).$$

3. Orthogonally project the full vector of ratings to $[-M, M]^n \cap \{x' \in \mathbb{R}^n \mid \sum_k x_k' = 0\}$.

Let $X_k^t$ denote the rating of Player $k$ at time $t$, and $\pi$ the equilibrium distribution on $\mathbb{R}^n$. Denote by

$$A_k^{t,T} := \tfrac{1}{t} \sum_{s=T}^{T+t-1} X_k^s \quad \text{for} \quad k \in [n] \quad \text{and} \quad t, T > 0$$

the *time-averaged ratings*. We typically start from $X^0 = (0, ..., 0)$. The (deterministic) time $T$ is a *burn-in phase*, which allows the Elo ratings to get 'near' the true skills, after which we start averaging.

*Remark* 2.2. The projection step ensures the Elo ratings do not become too large. It is required for our analysis, but not usually implemented in practice. Indeed, our experiments, discussed in §4, suggest that, as long as the step-size $\eta$ is small enough, Elo ratings remain bounded. Algorithms which estimate BTL parameters typically require some sort of projection [20] or regularisation [27]. A simple and efficient algorithm to realise the orthogonal projection is presented in the Appendix.

We show, for a suitable choice of parameters $t$, $T$ and $\eta$, that the time-averages are concentrated around the true ratings. In other words, we can use Elo to obtain an MCMC estimate of the true ratings.

Elo ratings in equilibrium are, in general, a *biased* estimator of the true ratings: i.e., $\mathbb{E}_\pi[X_k^0] \neq \rho_k$. Hence, there will be both a *bias* and an *error* term in our MCMC-type estimate of $\|A^{t,T} - \rho\|_2$.

The MCMC convergence rate depends on a *spectral gap* $\lambda_q$. This parameter quantifies how fast *local* information about the relative strengths of two players is propagated to the rest of the ratings. Similar parameters appear in most of the related literature, e.g., [37, 27].

**Definition 2.3** (Spectral Gap). Let $q$ be a distribution on unordered pairs in $[n]$. Define $q_{i,j} := q_{\{i,j\}}$, $d_{i,j} := \mathbf{1}\{i = j\} \sum_k q_{i,k}$ and $\Delta_{i,j} := d_{i,j} - q_{i,j}$ for $i, j \in [n]$. Let $\lambda_q$ denote the *spectral gap* (second smallest eigenvalue) of the *Laplacian* $\Delta$. Always, $\Delta\mathbf{1} = \mathbf{0}$ and $\Delta$ is positive semi-definite.

*Remark* 2.4. Equivalently, $\lambda_q$ is the spectral gap of the continuous-time Markov chain on $[n]$ with transition rates $q_{i,j} = q_{\{i,j\}}$ for $i, j \in [n]$. Note the scaling: $\sum_{i,j} q_{i,j} = 2$. So, the typical time until the continuous-time chain jumps is order $n$, not order 1. This implies $\lambda_q \leq 4/n$; see Lemma B.2.

We assume $\lambda_q > 0$. This holds unless there exists a non-empty subset $S \subsetneq [n]$ with $\sum_{i \in S, j \notin S} q_{i,j} = 0$—i.e., players in $S$ never play those in $S^c$. This makes estimation of the ratings impossible.

Our main result measures the disparity between the time-averaged Elo ratings and the true ratings.

**Theorem 2.5** (Convergence Rate). *Let $X \sim \mathrm{Elo}_M(q, \rho; \eta)$, as in Definition 2.1. Let $C_1, C_2 < \infty$. Then, there exists a constant $C_0$, depending only on $(C_1, C_2)$, such that if*

$$\min\{\lambda_q, \eta, 1/t\} \geq n^{-C_1} \quad \text{and} \quad \min\{t, T\} \geq C_0 t_\star, \quad \text{where} \quad t_\star := e^{2M}\eta^{-1}\lambda_q^{-1}\log n,$$

*then*

$$\mathbb{P}\left[\tfrac{1}{n}\|A^{t,T} - \rho\|_2^2 \leq \frac{C_0 e^{4M}}{\lambda_q n}\left(\eta + \frac{(\log n)^2}{\lambda_q t}\right)\right] \geq 1 - n^{-C_2}.$$

*In particular, if $\eta \asymp (\log n)^2/(\lambda_q t)$, then*

$$\tfrac{1}{n}\|A^{t,T} - \rho\|_2^2 \lesssim \frac{e^{4M}(\log n)^2}{\lambda_q n}\frac{1}{\lambda_q t} \quad \text{whp} \quad \text{as} \quad n \to \infty.$$

*Remark* 2.6. Ideally, $\lambda_q n = \tilde{\Omega}(1)$; e.g., if $q$ is uniform over the edges of an *expander graph*, then $\lambda_q n \asymp 1$. In this case, we view $(\log n)^2/(\lambda_q n) = \tilde{O}(1)$ as the error's *pre-factor* and $1/(\lambda_q t)$ as the *(squared) convergence rate*. Also, we can then choose $\eta = \tilde{\Omega}(1)$ and obtain non-trivial convergence results. This choice of $\eta$ is comparable to that used in practice. Moreover, on average, only $\tilde{O}(1)$ games *per player* are need to be observed to guarantee good approximation of the ratings.

The $\eta$ term arises from the average bias $\tfrac{1}{n}\|\mathbb{E}_{X \sim \pi}[X] - \rho\|_2^2$, which is non-zero in general. An estimate on the average variance $\tfrac{1}{n}\sum_k \mathbb{V}\mathrm{ar}_{X \sim \pi}[X_k]$ is necessary to obtain the MCMC convergence rate.

**Theorem 2.7** (Bias and Variance). *If $\pi$ is the equilibrium distribution of* $\text{Elo}_M(q, \rho; \eta)$, *then*

$$\frac{1}{n}\| \mathbb{E}_{X \sim \pi}[X] - \rho \|_2^2 \leq 4e^{4M}\eta/(\lambda_q n) \quad and \quad \frac{1}{n}\sum_k \mathbb{V}\text{ar}_{X \sim \pi}[X_k] \leq 4e^{2M}\eta/(\lambda_q n).$$

Despite Elo ratings being biased, the estimated probability a player wins their next match is not:

$$\sum_{j \in [n]} q_{i,j}\, \mathbb{E}_{X \sim \pi}[\sigma(X_i - X_j)] = \sum_{j \in [n]} q_{i,j}\sigma(\rho_i - \rho_j) \quad \text{for all} \quad i \in [n].$$

This holds when no projection step is performed as part of the Elo update (equivalently, $M := +\infty$).

The *mixing time* is more delicate, as the chain makes deterministic-size discrete jumps, but its equilibrium distribution is continuous and, therefore, the chain does not converge in total variation. Instead, we measure convergence in the *Wasserstein*, also known as *transportation*, distance.

**Theorem 2.8** (Contraction). *If $X, Y \sim \text{Elo}_M(q, \rho; \eta)$, then there exists a step-by-step coupling with*

$$\mathbb{E}_{(x,y)}[\|X^t - Y^t\|_2^2] \leq (1 - \kappa)^t \|x - y\|_2^2 \quad where \quad \kappa := \tfrac{1}{8}e^{-2M}\eta\lambda.$$

Markov chains satisfying the contraction property (i.e., *positively curved*) satisfy powerful concentration inequalities, developed particularly by Joulin and Ollivier [22, 23, 34, 33, 24]. The idea is that if $\|X^0 - Y^0\|_2 = D^0$, then $D^t := \|X^t - Y^t\|_2$ is small when $t \asymp \kappa^{-1} \log D^0$. For us, $D^0 \leq 2Mn$, leading to $t_\star \asymp \kappa^{-1} \log n$. In the reversible case, this can provide bounds, e.g., on the spectral gap.

The approach is particularly applicable to Markov chains on *finite* metric spaces, such as graphs. The set-up of finite graphs was analysed by Bubley and Dyer [12] under the name *path coupling*. In this case, $\mathbb{E}[D^t] \leq \frac{1}{2}$ implies $\mathbb{P}[X^t = Y^t] \geq \frac{1}{2}$, since the minimum graph distance between $x \neq y$ is 1.

Our underlying metric space is $(\mathbb{R}^n, \|\cdot\|_2)$, however. Inequalities for this more general set-up have been developed, but they often give weaker bounds. In our case, they lose a factor $1/(\eta\lambda_q) \gtrsim n$ in the convergence rate. Morally, though, the exponential convergence at rate $\kappa$ still implies that $1/\kappa$ is the correct timescale for mixing and concentration, perhaps up to some logarithmic factors. Establishing this rigorously is the most challenging part of our work from a technical point of view.

## 2.2 Comparison with Related Work

In this section, we compare the convergence rate of Elo given by Theorem 2.5 against the state of the art for BTL estimation. We highlight, once again, that previous work focusses on the problem of *offline* estimation, whilst Elo works in the more challenging online setting. Nevertheless, Elo is able to match the state-of-the-art algorithms in the offline setting for a wide range of parameters.

Our results imply that Elo provides an estimator $\hat{\rho}$ of $\rho$ with $\|\hat{\rho} - \rho\|_2^2 \lesssim e^{4M}(\log n)^2/(\lambda_q^2 t)$ whp. This matches, up to a log factor, the results by Hajek, Oh and Xu [20], who prove that the MLE constrained on $[-M, M]^n \cap \{x \in \mathbb{R}^n \mid \sum_k x_k = 0\}$, $\rho_{\text{MLC}}$, satisfies $\|\rho_{\text{MLC}} - \rho\|_2^2 \lesssim e^{8M} \log n/(\lambda_q^2 t)$. The dependency on $\lambda_q$ has been improved by Shah, Balakrishnan and Bradley [37], who show $\|\rho_{\text{MLC}} - \rho\|_2^2 \lesssim ne^{8M} \log n/(\lambda_q t)$. Since $n/\lambda_q \geq 1/4$, this is at least as good as our result, and potentially better for certain choices of $q$. However, up to log factors, our result matches theirs when $q$ corresponds to sampling edges of an expander graph—e.g., a complete graph, or an Erdős–Rényi graph with parameter $p \gg (\log n)/n$. Our result improves the constant in front of $M$.

More recently, Li, Shrotriya and Rinaldo [27] proved a regularised version of the MLE, $\rho_{\text{MLR}}$, achieves $\|\rho_{\text{MLR}} - \rho\|_2^2 \lesssim e^{2M_E}\delta/(\lambda_q^2 t)$, where $\delta$ is the ratio between the maximum and average degree of the comparison graph $G = ([n], E)$ with $E = \{\{i, j\} \mid q_{i,j} > 0\}$ and $M_E = \max_{i,j:q_{i,j}>0} |\rho_i - \rho_j| \leq 2\max_k |\rho_k|$. A version of our analysis also applies with $M_E$, instead of $M$, but we are not able to prove $\max_{i,j:q_{i,j}>0} |X_i^t - X_j^t| < M_E + 1$ holds for polynomially long. Notice that their result is weaker when the comparison graph is relatively sparse but has a few high degree nodes. Moreover, they require each player to play the same number of matches; i.e., $q_{i,j} = 1/|E|$ for $\{i, j\} \in E$. Both limitations are particularly problematic in the context of tournament design discussed in §3. On the other hand, they obtain $\ell_\infty$ error bounds too. We refer to [27] for further discussion of previous work.

## 2.3 Outline of Proof

We now highlight the key steps in estimating the MCMC-type error $\frac{1}{n}\|A^{t,T} - \rho\|_2^2$ in Theorem 2.5, where $A^{t,T}$ is the $t$-step time average of the ratings $X$ after a burn-in phase of length $T$. We use

$$\|A^{t,T} - \rho\|_2 \leq \|A^{t,T} - \mathbb{E}_\pi[X^0]\|_2 + \|\mathbb{E}_\pi[X^0] - \rho\|_2,$$

bounding these *error* and *bias* terms separately, via Theorems 2.8 and 2.7, respectively. Their proofs rely on estimating the change in $\ell_2$ norm a single step, and using the Lipschitz property of the sigmoid function $\sigma$, along with some other careful manipulations. The full proofs are given in the appendix.

The primary challenge is to leverage the positive curvature result of Theorem 2.8 to deduce the required concentration inequality in Theorem 2.5. As noted at the end of §2.1, positive curvature is well-suited to this goal, but the specifics of our set-up cause significant difficulties. Particularly, the aforementioned results reliant only on curvature are not sufficiently strong for our purposes.

A further complication is that Elo is a *non*-reversible Markov chain. There is a general understanding, or perhaps belief, in the MCMC community that non-reversibility can improve concentration results. This has lead to many strategies being proposed, such as non-backtracking, lifting, or zig-zag; see, e.g., [14, 16, 30, 8, 25]. Even an alternative definition of the spectral gap has been proposed [13]; this gap controls convergence of empirical averages, rather than convergence to equilibrium.

Unfortunately, the majority of general concentration results based on spectral properties [19, 26, 35, 36] are actually *worse* in the non-reversible set-up: they bound the convergence rates by those of the multiplicative or additive reversibilisation. These can be hard to estimate, needing detailed knowledge of the equilibrium distribution, and the resulting bounds are often crude.

Finally, we mention that bounds on the total-variation mixing time can be leveraged to provide concentration bounds; see, particularly, [15, 35]. However, Elo does not converge in total variation! Indeed, $X^t$ is finitely supported, on at most $(2n)^t$ states, given $X^0$, but $\pi$ is continuous. These $(2n)^t$ states can be used to distinguish $X^t$ and $\pi$. Nevertheless, it is this mixing-time approach that we adapt.

The contraction property implies that two realisations $X$ and $Y$ can be coupled so that their expected relative distance decreases exponentially, with rate $\kappa$. This is ideal for using the coupling approach to mixing. However, it is not possible to guarantee that $X^t = Y^t$ at some point, due to the chain's finite support. Instead, we analyse a *noisy version*: after each update, some (continuous) noise is added to the ratings. This must be added carefully to preserve the contraction in Theorem 2.8.

Denote the noisy versions $U$ and $V$. Once $U$ and $V$ are sufficiently close, the additive noise can be used to couple them exactly, to get $U^t = V^t$. The size of the noise must be balanced: too small and this final coupling step is too difficult; too big and $U$ can no longer be compared with $X$.

We delve deeper into this noisy approximation, explaining what noise to add, how it accumulates and what the resulting mixing time is. The interaction of the noise with the projection step is delicate and technical; we omit it from the description here, but it should be kept in the back of the mind.

*Noisy Version & Error.* We consider the following noisy version $U$ of the Elo process $X$. Let $\delta > 0$.

1. Draw $U^{t+1/2}$ according to an *uncapped* Elo step started from $U^t$—i.e., as if $M = \infty$. Suppose that Players $i$ and $j$ were chosen—i.e., $\{k \mid U_k^{t+1/2} \neq U_k^t\} = \{i, j\}$.

2. Draw $\tilde{U}_i, \tilde{U}_j \sim^{\text{iid}} \text{Unif}([-\sqrt{\delta}, +\sqrt{\delta}])$ independently and set $\tilde{U}_k := 0$ for $k \notin \{i, j\}$. Set

$$U_k^{t+1} := U_k^{t+1/2} + \tilde{U}_k \quad \text{for} \quad k \in [n].$$

This *does not* preserve the zero-sum property, as the additive noise is independent. The independence is *crucial* later to allow two version to coalesce. We need to control the cumulation until that point.

We control the difference between $X$ and $U$ via the natural coupling: pick the same pair of players to play, and observe the same result; sample the noise independently. Careful computation gives

$$\|X^{t+1} - U^{t+1}\|_1 \leq \|X^{t+1/2} - U^{t+1/2}\|_1 + 2\sqrt{\delta} \leq \|X^t - U^t\|_1 + 2\sqrt{\delta} \leq ... \leq 2t\sqrt{\delta}.$$

The second inequality actually says that Elo is non-negatively curved in $\ell_1$. Iterating this,

$$\left\| \tfrac{1}{t} \sum_{s=0}^{t-1} X^s - \tfrac{1}{t} \sum_{s=0}^{t-1} U^s \right\|_1 \leq \tfrac{1}{t} \sum_{s=0}^{t-1} \|X^s - U^s\|_1 \leq t\sqrt{\delta} \quad \text{deterministically.} \qquad \triangle$$

*Curvature.* We use the natural coupling between two noisy versions $U$ and $V$: choose the same players, observe the same result and add the same noise. Then, there is still rate-$\kappa$ contraction:

$$\mathbb{E}_{u^0, v^0}[\|U^1 - V^1\|_2] \leq \mathbb{E}_{u^0, v^0}[\|U^{1/2} - V^{1/2}\|_2] \leq (1 - \kappa)\|u^0 - v^0\|_2. \qquad \triangle$$

*Mixing Time.* We bound the total-variation mixing time of the noisy chain via the coupling method:

$$\left\| \mathbb{P}_{u^0}[U^t \in \cdot] - \mathbb{P}_{v^0}[V^t \in \cdot] \right\|_{\mathrm{TV}} \leq \mathbb{P}_{(u^0, v^0)}[U^t \neq V^t].$$

First, we burn-in using the natural coupling, then use a new coupling which exploits the noise.

We start with the natural coupling as given above. Using the rate-$\kappa$ curvature,

$$\mathbb{E}[\|U^t - V^t\|_2] \leq (1 - \kappa)^t \|U^0 - V^0\|_2 \leq 2Mne^{-\kappa t} \leq \delta^2 \quad \text{if} \quad t \geq t_\delta := \kappa^{-1} \log(2Mn/\delta^2).$$

Hence,

$$\mathbb{P}[\|U^t - V^t\|_\infty > \delta] \leq \mathbb{P}[\|U^t - V^t\|_2 > \delta] \leq \delta \quad \text{if} \quad t \geq t_\delta.$$

Once the absolute difference $|U_k^{t_\delta} - V_k^{t_\delta}|$ are at most $\delta$ for all $k$, the additive noise, which is order $\sqrt{\delta}$, dominates the change in difference after a single Elo step, which is order $\delta$. Thus, with complementary probability order $\delta/\sqrt{\delta} = \sqrt{\delta}$, we can couple the noise so that the rating of a player is the same in both $U$ and in $V$ after they play a game. Moreover, such a *successful* step preserves the $\ell_\infty$ bound of $\delta$.

The $\ell_\infty$ bound implies that all players are chosen within $t_\delta$ steps with probability at least $1 - \delta$. Hence, the probability of *not* successfully matching all ratings after $t_\delta$ steps is at most order $\delta + \sqrt{\delta} t_\delta$.

Combining these two bounds gives a bound of order $\delta + \sqrt{\delta} t_\delta$ on the total-variation distance at time $2t_\delta$. The polynomial-growth/decay assumptions in the theorem allow us to choose $\delta$ to be an appropriate inverse polynomial (in $n$) and obtain $\delta + \sqrt{\delta} t_\delta \ll 1$ and $t_\delta \asymp \kappa^{-1} \log n \asymp t_\star$. $\triangle$

The above mixing-time bound allows us to establish concentration of time-averages of the noisy Elo ratings, using results from [35]. Particularly, under certain assumptions, if $Z$ is a Markov chain with equilibrium distribution $\pi$ and mixing time $t_\star$, then

$$-\log \mathbb{P}_\pi\left[\left|\tfrac{1}{t} \sum_{s=0}^{t-1} f(Z^s) - \pi_f\right| \geq \zeta\right] \gtrsim \sigma_f^{-2} \zeta^2 t/t_\star,$$

where $\pi_f = \mathbb{E}_\pi[f]$ and $\sigma_f^2 := \mathbb{V}\mathrm{ar}_\pi[f]$. Notice that this requires $Z^0 \sim \pi$, which we do not impose; this requirement can be circumnavigated by using a burn-in, again comparing with a noisy version.

We want to take $f := f_k$ to be the projection onto the $k$-th coordinate, which is the $k$-th player's rating, for each $k$, then do a union bound over the $n$ players. This motivates taking

$$\zeta_k^2 \asymp \sigma_{f_k}^2 t_\star \frac{\log n}{t}, \quad \text{which satisfies} \quad \frac{1}{n} \sum_k \zeta_k^2 \asymp \frac{\eta}{\lambda_q n} \frac{\log n}{\lambda_q \eta} \frac{\log n}{t} = \frac{1}{\lambda_q n} \frac{(\log n)^2}{\lambda_q t},$$

using Theorem 2.7. This is the decay rate required in Theorem 2.5, but for the *noisy* version.

We need to compare the noisy version with the original. We do this via the estimate established earlier:

$$\left\| \tfrac{1}{t} \sum_{s=0}^{t-1} X^s - \tfrac{1}{t} \sum_{s=0}^{t-1} U^s \right\|_1 \leq \tfrac{1}{t} \sum_{s=0}^{t-1} \|X^s - U^s\|_1 \leq t\sqrt{\delta} \quad \text{deterministically.}$$

We are taking $t$ to be polynomial in $n$, and can choose $\delta^{-1}$ to be a sufficiently large polynomial so that this accumulated error is small. Care must be taken to to make all this rigorous, but once it is done, we are able to deduce the concentration result for the original Elo process.

## 3 Tournament Design

### 3.1 Our Results

In this section, we assume we are given a *comparison graph* $G = ([n], E)$, where edge $\{i, j\} \in E$ indicates that Players $i$ and $j$ are able to play against one another. We want to construct a distribution $q$ over edges $E$ of $G$ so that Elo can most efficiently approximate the true ratings of the players.

The decay rate in Theorem 2.5 is governed by the spectral gap $\lambda_q$. So, we want to maximise $\lambda_q$. Let

$$\lambda_{\mathrm{cts}}^\star := \sup\{\lambda_q \mid q \in [0, 1]^E, \ \textstyle\sum_{e \in E} q_e = 1\}.$$

This equals the largest spectral gap achievable by a *continuous-time* Markov chain on $[n]$ with transitions only across edges of $G$ and average jump-rate $1/n$. It is the *fastest-mixing Markov chain* problem, introduced by Sun et al. [39], and can be formulated as a semidefinite program. Olesker-Taylor and Zanetti [32] recently proved that $\lambda_{\mathrm{cts}}^\star n \gtrsim 1/(\mathrm{diam}\, G)^2$. This implies the following.

**Corollary 3.1** (Optimised $q$). *Suppose that the comparison graph $G = ([n], E)$ is given. In the set-up of Theorem 2.5, there exists a distribution $q$ on the edges $E$ such that*

$$\tfrac{1}{n}\|A^{t,T} - \rho\|_2^2 \lesssim n(\operatorname{diam} G)^2 (\log n)^2 / t \quad \text{whp} \quad \text{if} \quad \eta \lesssim n(\operatorname{diam} G)^2 (\log n)^2 / t.$$

This choice of probabilities $q$ can improve drastically over uniform weights—as used by [27]. E.g., if $G$ consists of two cliques connected by $k$ edges, then $\lambda_q n \lesssim k/n^2$ when $q_e := 1/|E|$ is uniform over $E$, whilst the optimal $\lambda_{\text{cts}}^\star n \asymp 1$ [32]. As a consequence, with the optimal choice of $q$, only $\tilde{O}(n)$ total matches need to be played to obtain a good approximation of the true ratings, compared with $\tilde{O}(n^3/k)$ for a uniform $q$. Here, $\tilde{O}(\cdot)$ indicates the asymptotic order up to logarithmic factors. This demonstrates the power of being able to *choose* $q$, given $G$.

Elo naturally parallelises: if two games consist of *disjoint* pairs of players, then the Elo update resulting from one is independent of the result of the other. Hence, if we wish to minimise the number of *rounds* (i.e., sets of games that can be played in parallel), rather than *games*, we should consider a distribution $\tilde{q}$ on the set $\mathcal{M}$ of *matchings* of the graph $G = ([n], E)$—i.e., collections of disjoint edges.

**Definition 3.2** (Parallel Elo). Let $\tilde{q}$ be a distribution on $\mathcal{M}$. A single step of $\operatorname{ParElo}_M(\tilde{q}, \rho; \eta)$ first selects a matching $S \subseteq E$ according to $\tilde{q}$ and applies the Elo update (with scale $\eta$) to each pair in $S$. Then, the resulting vector is orthogonally projected back to zero-sum vectors in $[-M, +M]^n$.

Our analysis is robust enough to handle the parallel case, obtaining convergence rate $1/(\lambda_q t)$, where now $q_e$ is the marginal probability that edge $e \in E$ appears in the matching:

$$q_e := \sum_{S \in \mathcal{M} : e \in S} \tilde{q}_S.$$

**Theorem 3.3** (Parallel). *Let $X \sim \operatorname{ParElo}_M(\tilde{q}, \rho; \eta)$. Then, under the conditions of Theorem 2.5,*

$$\tfrac{1}{n}\|A^{t,T} - \rho\|_2^2 \lesssim \frac{e^{4M}(\log n)^2}{\lambda_q n / N} \frac{1}{\lambda_q t} \quad \text{whp} \quad \text{if} \quad \eta \asymp \frac{(\log n)^2}{\lambda_q t},$$

*where $N := \sum_{e \in E} q_e$ is the mean size of the matching. To emphasise, here, $t$ and $T$ count rounds.*

*Remark* 3.4. It can be shown that this factor $N$ is needed in the pre-factor via a time-change analysis.

It is natural to optimise $\lambda_q$ over $q$ which can arise as the marginals of a distribution $\tilde{q}$ on $\mathcal{M}$. Clearly,

$$q_k := \sum_{e \in E : k \in e} q_e \leq 1 \quad \text{for all} \quad k \in [n]$$

for any such $q$. Thus, the matrix $Q = (q_{i,j})_{i,j \in V}$ is substochastic, so $\lambda_q$ corresponds to the spectral gap of the *discrete-time* Markov chain on $G = ([n], E)$ with weights $q_{i,j} = q_{\{i,j\}}$. Let

$$\lambda_{\text{disc}}^\star := \sup\{\lambda_q \mid q \in [0,1]^E, \ \max_{k \in [n]} q_k \leq 1\}.$$

This is the optimal spectral gap of a *discrete-time* Markov chain on $[n]$ with transitions allowed only across edges of $G$ and uniform stationary distribution. Again, $\lambda_{\text{disc}}^\star$ can be formulated as a semidefinite program [10] and is related to the *vertex conductance* via a Cheeger-type inequality [32].

We show that a distribution $\tilde{q}$ over matchings with $\lambda_q \geq \frac{1}{3}\lambda_{\text{disc}}^\star$ can be found by decomposing the substochastic $Q$ into a convex combination of permutation matrices using the Birkhoff–von-Neumann theorem, followed by decomposing each permutation into disjoint cycles.

**Corollary 3.5** (Optimised $\tilde{q}$). *Suppose that the comparison graph $G = ([n], E)$ is given. In the set-up of Theorem 3.3, there exists a distribution $\tilde{q}$ on matchings $\mathcal{M}$ such that*

$$\tfrac{1}{n}\|A^{t,T} - \rho\|_2^2 \lesssim \frac{e^{4M}(\log n)^2}{\lambda_{\text{disc}}^\star n / N} \frac{1}{\lambda_{\text{disc}}^\star t} \quad \text{whp} \quad \text{if} \quad \eta \asymp \frac{(\log n)^2}{\lambda_{\text{disc}}^\star t}.$$

In many examples, such as if $G = ([n], E)$ is an expander, but also if $G$ is a cycle, then $\lambda_{\text{disc}}^\star \asymp n\lambda_{\text{cts}}^\star$. In this case, parallel Elo really is as good, up to constants, as $n$ steps of the original ('series') Elo. (Recall that $N = \sum_e q_e = 1$ is required for $\lambda_{\text{cts}}^\star$, but that $N \asymp n$ is possible for $\lambda_{\text{disc}}^\star$.)

Other times, there is already a continuous-time chain, with average jump-rate 1, which has spectral gap order 1; e.g., two cliques connected by a single edge. In this case, the optimal parallel version gives no real improvement, even measured by *rounds*, over the optimally weighted series version.

## 3.2 Comparison with Related Work

The problem of designing an efficient tournament graph is related to *active ranking* [21]. Active ranking, however, allows one to choose which matches to schedule next *after* observing the results of some matches. For example, Yan et al. [40] propose an algorithm to identify the *most informative* pair of players given previous outcomes, and obtain regret bounds between Elo and the true ratings when matchups are scheduled according to their algorithm. In contrast, we are interested in designing a probability distribution over matches in an *offline* manner, without the possibility of changing such distribution after observing some results.

This problem has been considered by Li, Shrotriya and Rinaldo [27]. They discuss a divide-and-conquer strategy that essentially requires oversampling edges across *bottlenecks* in the graph. The drawback of this strategy is that it requires partitioning the graph into well-connected pieces, which is a non-trivial task itself. Furthermore, [27] does not provide explicit bounds on the sample complexity that can be obtained in this way, besides discussing a few examples where the bottleneck is known.

Nonetheless, our approach shares some similarity with [27]: the fastest-mixing Markov chain implicitly up-weights edges across bottlenecks. The main advantage, however, is that it provides the *optimal* spectral gap allowed by a graph topology, which is related to the diameter of the graph [32]. Moreover, both $\lambda_{cts}^\star$ and $\lambda_{disc}^\star$ can be formulated as SDPs, for which fast (polynomial-time) solvers exist. We remark, however, that this construction might require certain nodes to play an overwhelmingly large number of matches: this would be problematic for the results of [27]. As far as we know, minimising the number of 'parallel rounds' has not been considered before.

# 4 Experimental Results

We close with discussion of some specific examples and experimental results. Additional experiments are discussed in the Appendix. We start by considering *dumbbell* tournament graphs consisting of two cliques of $n/2 = 20$ vertices connected by a matching of $k \in \{1, 20\}$ edges. For each graph, we perform Elo simulations where the match-ups between players are sampled according to the following probability distributions.

1. The *uniform* distribution $q_\mathcal{U}$ over the edges of the graph.
2. The *optimal sequential* $q_{seq}^\star$ derived from the fastest-mixing continuous-time Markov chain.
3. The distribution over matchings $q_{par}^\star$ derived from the fastest mixing discrete-time Markov chain, where multiple games are played in *parallel* in each *round*.

From §3, we know that $\lambda_{q_\mathcal{U}} \asymp k/n^3$, $\lambda_{q_{seq}^\star} \asymp 1/n$ and $\lambda_{q_{par}^\star} \asymp k/n$.

We sample the true ratings of the players according to independent Gaussians, with mean equal to 1 on one clique, mean 2 on the other, and standard deviation equal to 0.2 in both. This difference in average ratings between cliques simulates a scenario where the two cliques correspond to two different leagues of slightly different strength on average.

We perform Elo simulations initialising the Elo ratings at zero and setting $\eta = 0.1$. For each graph, we repeat each experiment ten times, each time sampling new true ratings. Experimental results are displayed in Figures 1 and 2. Simulations are repeated ten times: lines correspond to the average $\ell_2^2$-distance between time-averaged Elo ratings and the true ratings, divided by the number of players ($n = 40$). Shaded regions corresponds to 25–75 percentile over these ten trials. Cyan and blue lines correspond to the same experiments where we have sampled multiple games in parallel as described in §3; we display the decay of error wrt the total number of games played in cyan and wrt the number of parallel rounds in blue. The blue line is solid for the first $2 \cdot 10^4$ games (same number of games displayed in cyan).

We display the experimental results for $k = 1$, i.e., two cliques of 20 vertices connected by a single edge, in Figure 1. The experiments align with the theoretical results of §3: when $k = 1$, the spectral gap $\lambda_{q_{seq}^\star}$ of the optimal sequential distribution is of the same order of the spectral gap for our nearly parallel construction $\lambda_{q_{par}^\star}$. Indeed, if convergence is measured wrt the number of rounds, the two corresponding errors seems to decay at a similar same rate, with the parallel version being slightly better, but requiring more than ten times the total number of games. As predicted by the theory, both distributions result in much faster convergence than the uniform distribution. If we measure the total

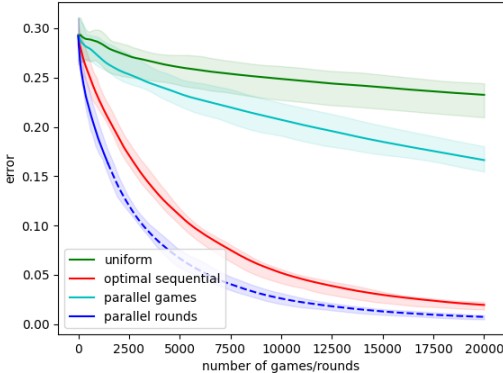
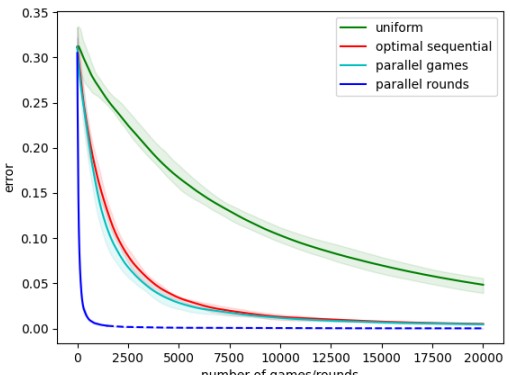

Figure 1: Elo simulation results for a dumbbell graph with one edge between two cliques of 20 vertices. Match-ups are sampled from three different probability distributions.

Figure 2: Elo simulation results for a dumbbell graph with a perfect matching of 20 edges between two cliques of 20 vertices.

number of games rather than rounds, the parallel version still results in a faster convergence than the uniform distribution, but not overwhelmingly so.

We now consider a dumbbell graph with $k = n/2$, i.e., two cliques connected by a perfect matching. In this case, the fastest discrete- and continuous-time Markov chains have the same order-1 spectral gap. However, since the spectral gap for sequential Elo is then rescaled by a factor of $1/n$, to achieve convergence, we expect the number of *rounds* for the parallel version to be much smaller than the number of *games* required by the sequential one; convergence should instead happen at the same rate when measured in the total number of games. This is clearly shown in Figure 2. The uniform distribution performs much worse, which we would expect from its order-$1/n^2$ spectral gap. In the optimal parallel and sequential distributions, the probability to sample an edge from the bottleneck or from inside the cliques is balanced, while the uniform distribution oversamples edges inside the cliques. Experiments for the intermediate case of $k \in \{5, 10\}$ are discussed in the Appendix.

We end this section by discussing experimental results concerning the maximum rating reached by Elo. Figure 3 displays the behaviour of the largest Elo rating in absolute value, where pairs of players are selected uniformly at random (i.e., the underlying graph is a complete graph). The true ratings are sampled uniformly at random in $[-1, 1]$. The initial Elo ratings are set equal to zero. We simulate up to 50000 matches for a number of players that goes from 100 to 1000. We observe that the maximum rating is always below 1.75, corroborating our belief that, in many scenarios, the maximum Elo rating does not diverge for a long time. Further experiments and discussions are given in the Appendix.

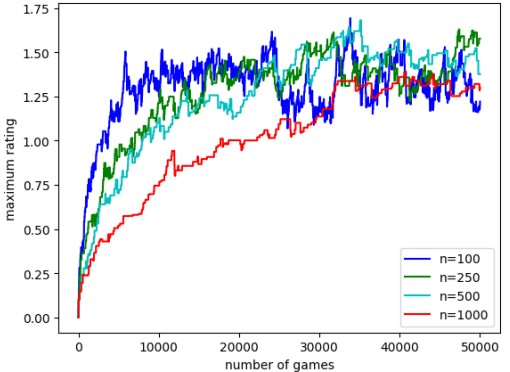

Figure 3: Largest Elo rating in absolute value for a complete graph of varying size. True ratings are uniformly distributed in $[-1, 1]$.

## 5   Conclusion and Open Problems

Our work is a first step towards establishing the theoretical foundations of the Elo rating system, a popular ranking method in sports analytics. In particular, our main contribution is an analysis of Elo under the BTL model, establishing convergence results competitive with the state of the art.

There are several questions prompted by our work. First, from a technical point of view, we would like to control the maximal Elo rating. This is necessary to understand when we can remove the projection step in our definition of the Elo Markov chain, and make our theoretical results more aligned with practice.

Moreover, we would like to better understand the shape of the stationary distribution of Elo. This could help us, for example, obtain bounds on the rate of convergence in $\ell_\infty$.

Finally, a touted strength of the Elo rating system in practical applications is its ability to dynamically update the ratings in response to changes in players' skills. Can we model these changes in a way that allows us to prove Elo can keep track of them, and bound the corresponding mean squared error? Such questions are often studied in the statistics literature; see, e.g., the recent paper [18] for details.

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

# A    Experiments: Further Discussion and Figures

In this appendix we discuss additional experimental results. Code for our experiments is included in the supplementary material.

In Figure 4 we display further experiments related to dumbbell graphs. In particular, we consider graphs consisting of two cliques of 20 vertices connected by 5 (left) and 10 (right) edges arranged in a matching. The experimental setup is the same as the one discussed in §4. Notice how, the more edges are added to the bottleneck, the better the distribution optimised for parallel rounds performs. The optimal sequential distribution, instead, performs roughly the same: adding edges in the bottleneck doesn't really improve its spectral gap, which depends mainly on the (unchanged) diameter.

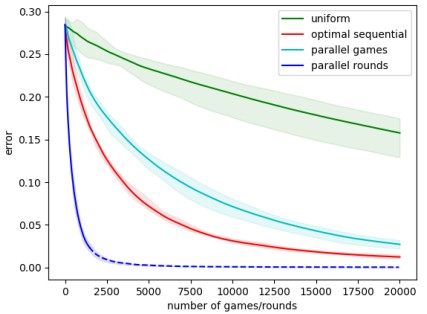 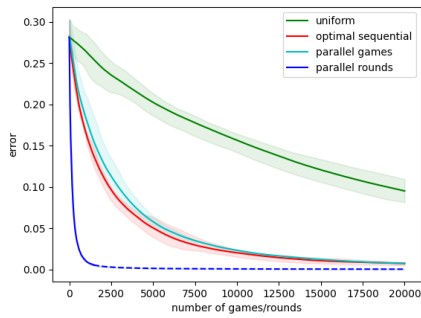

Figure 4: Elo simulation results for dumbbell graphs with $k = 5$ (left) and $k = 10$ (right) edges between two cliques of 20 vertices.

We also consider a *pyramidal* graph, which is constructed as follows. We first sample three Erdős–Rényi random graphs with size, resp., $n_1 = 64$, $n_2 = 32$ and $n_3 = 16$, and density $p = 1/2$. We then connect the first graph to the second and the second to the third with two sparse cuts (see Figure 5). This graph is constructed to loosely resemble the pyramidal structure of, e.g., national sport leagues.

We conduct Elo simulations with the same set-up as for dumbbell graphs discussed earlier. In particular, we repeat the simulations ten times, resampling each time the players' true ratings. The true ratings are sampled as follows: independent normal distributions of standard deviation 0.2 and mean 0 for the Erdős–Rényi at the bottom of the pyramid, mean 1 for the Erdős–Rényi in the middle,

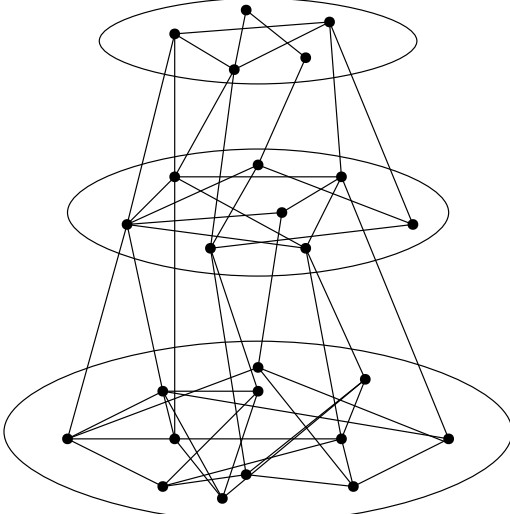
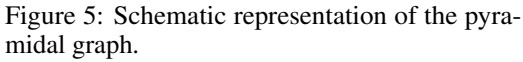

Figure 5: Schematic representation of the pyramidal graph.

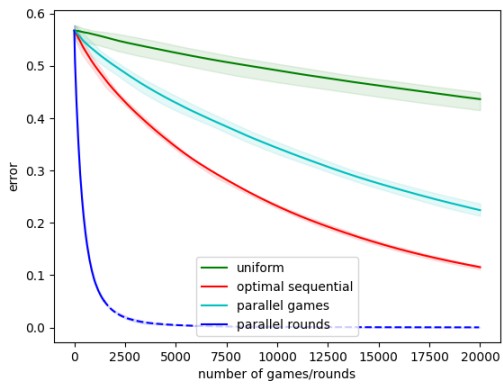

Figure 6: Elo simulation results for the pyramidal graph.

and mean 2 for the one at the top. This is, again, to loosely simulate the fact that sport leagues are characterised by stronger players/teams towards the top of the pyramid. Experimental results are shown in Figure 6. In particular, we observe that the rate of converge for the uniform distribution is much slower than for the optimal sequential one. This is, again, predicted by the results of §3: the two bottlenecks slow down the convergence in the uniform case; while the small diameter assures faster convergence for the optimal distribution by Corollary 3.1. The distribution optimised for parallel Elo, instead, guarantees very fast convergence when measured according to the number of rounds.

In Figure 7 we display experimental results for a topology corresponding to the giant component of an Erdős–Rényi random graph of $n = 100$ vertices and density $p = 0.02$. True ratings are distributed as independent standard Gaussians. This graph has reasonably good connectivity, so we expect the number of games required to converge to be roughly equal for all the sampling distributions considered, with a good scope for parallelisation. This is confirmed by the error plot. What is perhaps surprising is that the optimal sequential distribution actually performs slightly worse than the uniform one. This can be explained by the fact that the spectral gap measures the rate of convergence for the worst possible vector of ratings: if the ratings do not depend on the topology of the graph, like in this example, the inverse of the spectral gap is an overtly pessimistic upper bound. Indeed, the optimal sequential distribution tends to oversample nodes that are in a central position in the graph and undersample nodes at the periphery. This should make convergence faster because it allows faster movement of information between distant nodes, but in this specific example is not helpful: since ratings are sampled in a iid fashion, the faster movement of information globally is not very helpful and is upset by slower convergence for undersampled nodes.

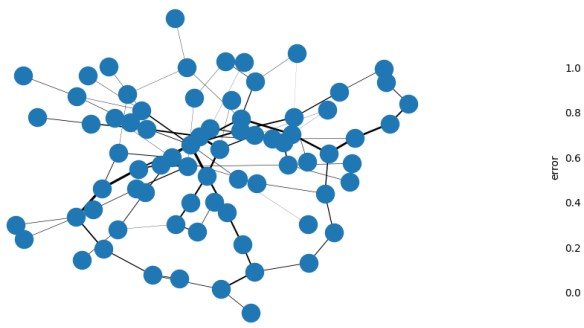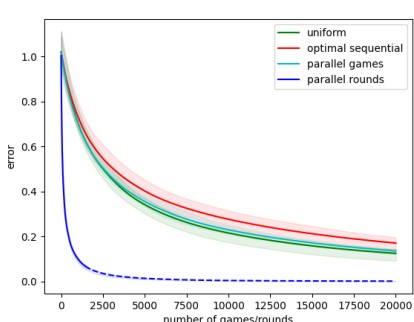

Figure 7: Left: topology of the giant component of an Erdős–Rényi random graph of $n = 100$ vertices and density $p = 0.02$; edges are reweighed according to the distribution corresponding to the fastest mixing continuous-time Markov chain. Right: Elo simulation results for the same graph.

We now present further experiments about the maximum Elo rating (in absolute value). Again, we sample true ratings independently and uniformly in $[-1, 1]$. In Figure 8, we present experiments for a path (left) and star (right) topology of varying size. In both cases, the largest Elo rating in absolute value remains smaller than twice the largest true rating. Notice that in the star graph the central node plays all the matches; this means the same player plays in total fifty-thousands matches without its value becoming particularly large.

Of course, these simulations are far from definitive: there are countless of ways in which to choose the true ratings and the graph topology. Our belief that these experiments are indicative of a more general behaviour is due to the peculiarities of the Elo rating systems. In particular, the Elo update offers diminishing returns: if the rating of $i$ is relatively much larger than the rating of $j$, if $i$ wins against $j$, the rating of $i$ won't increase by much. This is because the sigmoid function $\sigma$ approaches zero very fast.

Despite this, *proving* that the maximum rating cannot increase significantly appears much harder. This is due to the fact that the maximum rating is not a supermartingale: if all of the neighbours of the node $i$ with largest true rating have abnormally large rating, the rating of $i$ is likely to increase, no matter how large it already was. However, because the zero-sum property of the ratings, the neighbours of $i$ cannot maintain an abnormally large rating for too long. Indeed, it appears very unlikely that such balanced but "abnormal" configurations are ever reached.

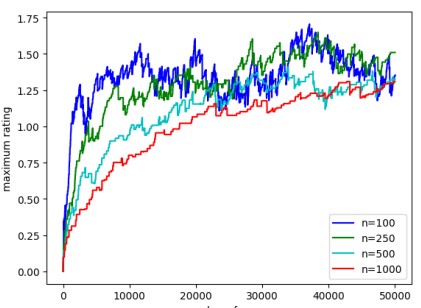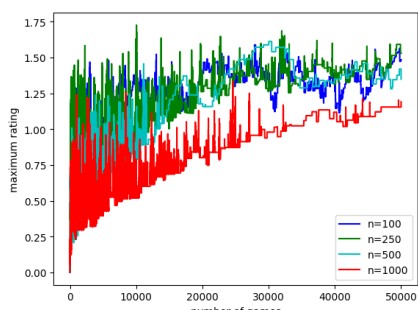

Figure 8: Behaviour of the largest Elo rating in absolute value for path (left) and star (right) graphs of varying size.

In the two-player case, the situation is simpler: if player $1$ has Elo rating $x_1$, then player $2$ has $x_2 = -x_1$, by the zero-sum nature. It is straightforward to check that the Elo ratings are biased towards the true skills: that is, if player $1$ has rating $x_1$ before a game, then their rating $x_1'$ after the game has $\mathbb{E}[|x_1' - \rho_1|] < |x_1 - \rho_1|$, if $\eta < \frac{1}{2}$.

Suppose that the same argument holds for $n > 2$ players: i.e., *typically*, Elo ratings are biased towards the true skills. Biased random walks on $\mathbb{R}$ have exponential tails. So, if $\eta \ll 1/\log n$, then it is super-polynomially unlikely that a given Elo rating will be more than 1 away from the corresponding true skill, in equilibrium. A union bound over the $n$ players allows us to deduce that the maximum Elo rating is at most 1 more than the maximum true skill, with probability at least $1 - 1/n^{10}$.

Unfortunately, we weren't able to make this argument formal for $n > 2$ players. We leave proving that indeed, with high probability, the maximum rating remains small for a large number of steps as an open problem.

# B  Convergence Proofs: Preliminaries

There are a few results which we use repeatedly throughout the proofs. We collect them here.

## B.1  Dirichlet Characterisation of the Spectral Gap

Recall that $\lambda_q$ is the spectral gap of the continuous-time Markov chain on $[n]$ with transition rates $(q_{i,j})_{i,j\in[n]}$. We repeatedly use the standard Dirichlet characterisation of the spectral gap.

**Proposition B.1** (Dirichlet Characterisation of the Spectral Gap). *The spectral gap $\lambda_q$ satisfies*

$$\lambda_q = \tfrac{1}{2} \min_{z\in\mathbb{R}^n\setminus\{0\}:z\perp 1} \sum_{i,j\in[n]} q_{i,j}(z_i - z_j)^2 / \|z\|_2^2.$$

*In particular, for all $z \in \mathbb{R}^n$ with $\sum_k z_k = 0$,*

$$\sum_{i,j\in[n]} q_{i,j}(z_i - z_j)^2 \geq 2\lambda_q \|z\|_2^2.$$

**Lemma B.2.** *Let $q_{i,j} \in [0,1]$ for all $i,j \in [n]$ with $\sum_{i,j} q_{i,j} = 2$. Let $\lambda_q$ denote the spectral gap of the continuous-time Markov chain on $[n]$ with transition rates $(q_{i,j})_{i,j\in[n]}$. Then,*

$$\lambda_q \leq 2/(n-1) \leq 4/n.$$

*Proof.* We apply the Dirichlet characterisation (Proposition B.1) with $z_i := \mathbf{1}\{i = k\} - \frac{1}{n}$. Then,

$$z \perp 1, \quad \|z\|_2^2 = (1 - 1/n)^2 + (n-1)/n^2 = 1 - 1/n = (n-1)/n$$

and

$$\begin{aligned}
\tfrac{1}{2} \sum_{i,j\in[n]} q_{i,j}(z_i - z_j)^2 / \|z\|_2^2 &= \tfrac{1}{2} \sum_{i,j\in[n]} q_{i,j}(\mathbf{1}\{i = k\} - \mathbf{1}\{j = k\})^2 \tfrac{n}{n-1} \\
&= \tfrac{1}{2}\tfrac{n}{n-1}\big(\sum_{j:j\neq k} q_{k,j} + \sum_{i:i\neq k} q_{i,k}\big) = \tfrac{n}{n-1} \sum_\ell q_{k,\ell}.
\end{aligned}$$

Choosing $k$ to minimise this final sum gives $\sum_\ell q_{k,\ell} \leq \frac{2}{n}$ since the average $\frac{1}{n}\sum_k \sum_\ell q_{k,\ell} = \frac{2}{n}$. So,

$$\lambda_q \leq \tfrac{n}{n-1} \cdot \tfrac{2}{n} = 2/(n-1) \leq 4/n. \qquad \square$$

## B.2 Capping the Ratings

Capping at $\pm M$ has the benefit of restricting the process remains in a compact set. The disadvantage, though, is that the Elo update is more complicated. It also biases the process compared with the original. However, as we discuss in Remark C.2 below, the original process *is already biased* in the sense that the expected rating in equilibrium *is not* the true rating: $\mathbb{E}_\pi[X] \neq \rho$.

The only property of the projection that we use is the following monotonicity result.

**Proposition B.3** (Projection Monotonicity). *Let $\Omega \subseteq \mathbb{R}^n$ be a closed, convex set. Define*

$$\Pi_\Omega(x) := \arg\min_{x' \in \Omega} \|x - x'\|_2 \quad for \quad x \in \mathbb{R}^n;$$

*that is, $\Pi_\Omega$ is the orthogonal projection to $\Omega$. Then,*

$$\|\Pi_\Omega(x) - \Pi_\Omega(y)\|_2 \leq \|x - y\|_2 \quad for \ all \quad x, y \in \mathbb{R}^n;$$

*that is, $\Pi_\Omega(x)$ is at least as close to $\Pi_\Omega(y)$ as $x$ is to $y$, in $\ell_2$.*

Versions of this result are well-known. An elementary proof can be found at [29]. We apply this with $\Omega := \{x \in \mathbb{R}^n \mid \|x\|_\infty \leq M, \sum_k x_k = 0\}$, which is convex. Our results apply whenever this monotonicity property holds.

The orthogonal projection $\Pi_\Omega(x)$ can be efficiently computed as follows. We first project $x$ to $\{y \in \mathbb{R}^n \mid \|y\|_\infty \leq M\}$. This can be done by simply replacing any $x_i < -M$ with $-M$ and any $x_i > M$ with $M$. Let $x'$ be the resulting vector. Notice that $x'$ might not satisfy the constraint $\sum_k x'_k = 0$: we need to subtract the displaced mass $\sum_k x'_k$ to the rest of the vector while minimising $\Pi_\Omega(x) := \arg\min_{x'' \in \Omega} \|x - x''\|_2$. Assume $\sum_k x'_k > 0$ (the symmetric case can be handled similarly). We set $x''_i = -M$ for all the coordinates $i$ such that $x'_i = -M$ and call $i$ *frozen*. To minimise $\|x - x''\|_2$, we want to distribute $-\sum_k x'_k$ to the unfrozen coordinates so that $x - x''$ is as balanced as possible in each coordinate (this can be imagined as a water-filling procedure). In doing so, however, we might make $x''_i = -M$ for a new, previously unfrozen, coordinate $i$ before we have subtracted all the displaced mass. If that happens, we simply freeze $i$ and proceed in distributing the remaining displaced mass to the unfrozen coordinates (until a new coordinate is frozen). We keep repeating the procedure since all the mass is allocated. Notice that we can allocate all the mass since we assumed $\sum_k x'_k > 0$. Moreover, the procedure lasts at most $n$ steps since at each step we freeze a new coordinate until we have completed constructing the desired vector.

# C Convergence Proofs: Bias and Variance

There is an inherent bias and variance in the Elo ratings. It would be natural to assume that the expected value of a player's rating in equilibrium is equal to their real skill. After all, at least in the two-player case, an individual step is always biased towards the real skill. However, even for two players, this is not true. In this section, we quantify the bias and the variance in equilibrium.

Throughout this section and the next, we denote

$$p_{i,j}(z) := \sigma(z_i - z_j) \quad for \quad z \in \mathbb{R}^n \quad and \quad i, j \in [n].$$

In particular, $p_{i,j}(\rho)$ is the *true* probability that Player $i$ beats $j$ and $p_{i,j}(x)$ is the *model* probability if the current ratings are $x$. Additionally, we split up the Elo step into the update and the projection:

- if the current vector is $X^t$, then let $X^{t+1/2}$ denote an *uncapped* step;
- then, $X^{t+1} = \Pi_M(X^{t+1/2})$, where $\Pi_M$ is the orthogonal projection.

We do not quantify the bias and variance player-by-player, but rather average over all players.

**Theorem C.1** (Bias and Variance; Theorem 2.7). *The following bias and variance estimates hold:*

$$\|\mathbb{E}_\pi[X^0] - \rho\|_2^2 \leq \mathbb{E}_\pi[\|X^0 - \rho\|_2^2] \leq 4e^{4M}\eta/\lambda_q;$$

$$\sum_k \mathbb{V}\mathrm{ar}_\pi[X_k^0] = \mathbb{E}_\pi[\|X^0 - \mathbb{E}_\pi[X^0]\|_2^2] \leq 4e^{2M}\eta/\lambda_q.$$

*Proof: Bias.* First and foremost, the Cauchy–Schwarz inequality gives

$$\|\mathbb{E}_\pi[X^0] - \rho\|_2^2 \leq \mathbb{E}_\pi[\|X^0 - \rho\|_2^2].$$

We start the system from equilibrium and use stationarity: $X^0 \sim \pi \implies X^1 \sim \pi$. Also, we can ignore the projection step, whilst still assuming that all vectors have $\ell_\infty$ norm at most $M$, due to the projection monotonicity of Proposition B.3: noting that $\|\rho\|_\infty \le M$, so $\Pi_M(\rho) = \rho$, we have

$$\|X^1 - \rho\|_2 = \|\Pi_M(X^{1/2}) - \pi_M(\rho)\|_2 \le \|X^{1/2} - \rho\|_2.$$

Write $\mathbb{E}_{x,\{i,j\}}[\cdot]$ to indicate that the pair $\{i,j\}$ is chosen in the first step and $X^0 = x$. Then,

$$
\begin{aligned}
\mathbb{E}_{x,\{i,j\}}\big[(X_i^{1/2} - \rho_i)^2\big] &= p_{i,j}(\rho)\big(x_i - \rho_i + \eta\big(1 - p_{i,j}(x)\big)\big)^2 + \big(1 - p_{i,j}(\rho)\big)\big(x_i - \rho_i - \eta p_{i,j}(x)\big)^2 \\
&= (x_i - \rho_i)^2 + \eta^2\big(p_{i,j}(\rho)\big(1 - p_{i,j}(x)\big)^2 + \big(1 - p_{i,j}(\rho)\big)p_{i,j}(x)^2\big) \\
&\quad + 2\eta\big(p_{i,j}(\rho)\big(1 - p_{i,j}(x)\big) - \big(1 - p_{i,j}(\rho)\big)p_{i,j}(x)\big)(x_i - \rho_i) \\
&\le (x_i - \rho_i)^2 + \eta^2 - 2\eta\big(p_{i,j}(x) - p_{i,j}(\rho)\big)(x_i - \rho_i).
\end{aligned}
$$

An analogous statement holds for $X_j^{1/2} - \rho_j$, with $i$ and $j$ swapped:

$$
\begin{aligned}
\mathbb{E}_{x,\{i,j\}}\big[(X_j^{1/2} - \rho_j)^2\big] &\le (x_j - \rho_j)^2 + \eta^2 - 2\eta\big(p_{j,i}(x) - p_{j,i}(\rho)\big)(x_j - \rho_j) \\
&= (x_j - \rho_j)^2 + \eta^2 + 2\eta\big(p_{i,j}(x) - p_{i,j}(\rho)\big)(x_j - \rho_j),
\end{aligned}
$$

using the fact that $p_{j,i}(\cdot) = 1 - p_{i,j}(\cdot)$. For $k \notin \{i,j\}$,

$$\mathbb{E}_{x,\{i,j\}}\big[(X_k^{1/2} - \rho_k)^2\big] = (x_k - \rho_k)^2.$$

Bringing these three cases (indices $k = i$, $k = j$ and $k \in [n] \setminus \{i,j\}$) together,

$$
\begin{aligned}
\mathbb{E}_{x,\{i,j\}}\big[\|X^{1/2} - \rho\|_2^2\big] &\le \|x - \rho\|_2^2 + 2\eta^2 - 2\eta\big(p_{i,j}(x) - p_{i,j}(\rho)\big)\big((x_i - \rho_i) - (x_j - \rho_j)\big) \\
&= \|x - \rho\|_2^2 + 2\eta^2 - 2\eta\big(p_{i,j}(x) - p_{i,j}(\rho)\big)\big((x_i - x_j) - (\rho_i - \rho_j)\big).
\end{aligned}
$$

Now, $p_{i,j}(x) - p_{i,j}(\rho) = \sigma(x_i - x_j) - \sigma(\rho_i - \rho_i)$ and $(x_i - x_j) - (\rho_i - \rho_j)$ have the same sign. Also,

$$
\begin{aligned}
|p_{i,j}(x) - p_{i,j}(\rho)| &= \big|\sigma\big(x_i - x_j\big) - \sigma\big(\rho_i - \rho_j\big)\big| \\
&\ge \min z \in [-4M, 4M]|\sigma'(z)| \cdot |(x_i - x_j) - (\rho_i - \rho_j)| \\
&\ge \tfrac{1}{4}e^{-4M}|(x_i - x_j) - (\rho_i - \rho_j)|
\end{aligned}
$$

since $x_i, x_j, \rho_i, \rho_j \in [-M, M]$, so $|(x_i - x_j) - (\rho_i - \rho_j)| \le 4M$ and $\sigma'(z) = \sigma(z)\big(1 - \sigma(z)\big)$. Hence,

$$
\begin{aligned}
\mathbb{E}_{x,\{i,j\}}\big[\|X^{1/2} - \rho\|_2^2\big] &\le \|x - \rho\|_2^2 + 2\eta^2 - \tfrac{1}{2}e^{-4M}\eta\big((x_i - x_j) - (\rho_i - \rho_j)\big)^2 \\
&= \|x - \rho\|_2^2 + 2\eta^2 - \tfrac{1}{2}e^{-4M}\eta\big((x_i - \rho_i) - (x_j - \rho_j)\big)^2.
\end{aligned}
$$

We now average this over $\{i,j\}$:

$$\mathbb{E}_x\big[\|X^{1/2} - \rho\|_2^2\big] \le \|x - \rho\|_2^2 + 2\eta^2 - \tfrac{1}{2}e^{-4M}\eta\sum_{i,j} q_{\{i,j\}}|(x_i - \rho_i) - (x_j - \rho_j)|^2.$$

Now, applying the Dirichlet characterisation of the spectral gap (Proposition B.1) with $z := x - \rho$,

$$\mathbb{E}_x\big[\|X^{1/2} - \rho\|_2^2\big] \le \|x - \rho\|_2^2 + 2\eta^2 - \tfrac{1}{2}e^{-4M}\eta\lambda_q\|x - \rho\|_2^2 = (1 - \tfrac{1}{2}e^{-4M}\eta\lambda_q)\|x - \rho\|_2^2 + 2\eta^2;$$

again, $\sum_k z_k = \sum_k x_k - \sum_k \rho_k = 0$. In particular, stationarity implies that

$$\mathbb{E}_\pi\big[\|X^0 - \rho\|_2^2\big] = \mathbb{E}_\pi\big[\|X^1 - \rho\|_2^2\big] \le \mathbb{E}_\pi\big[\|X^{1/2} - \rho\|_2^2\big] \le 4e^{4M}\eta/\lambda_q. \qquad \square$$

We perform a very similar calculation when bounding the *curvature*; see Definition D.2. This is the exponential contraction rate between a pair of systems $X$ and $Y$. The $(1 - \tfrac{1}{2}e^{-4M}\eta\lambda_q)$-factor above suggests curvature $\kappa \asymp e^{-4M}\eta\lambda_q$, which is indeed what we show in Proposition D.4.

We now turn to the variance, for which we use a similar approach. It is a little more technically challenging, using the slightly cumbersome *law of total variance*: for random variables $A$ and $B$,

$$\mathbb{V}\mathrm{ar}[B] = \mathbb{E}[\mathbb{V}\mathrm{ar}[B \mid A]] + \mathbb{V}\mathrm{ar}[\mathbb{E}[B \mid A]].$$

*Proof: Variance.* The sum of variances is the expectation of an $\ell_2$ distance:

$$\sum_k \mathbb{V}\mathrm{ar}[Z_k] = \sum_k \mathbb{E}\left[(Z_k - \mathbb{E}[Z_k])^2\right] = \mathbb{E}\left[\sum_k (Z_k - \mathbb{E}[Z_k])^2\right] = \mathbb{E}\left[\|Z - \mathbb{E}[Z]\|_2^2\right],$$

for any random variable $Z \in \mathbb{R}^n$. Also, for any $c \in \mathbb{R}^n$,

$$\mathbb{V}\mathrm{ar}[Z] = \mathbb{E}[(Z - \mathbb{E}[Z])^2] \leq \mathbb{E}[(Z - c)^2].$$

Applying this and the monotonicity of the orthogonal projection from Proposition B.3 gives

$$\begin{aligned}
\sum_k \mathbb{V}\mathrm{ar}[X_k^1] &= \mathbb{E}\left[\|X^1 - \mathbb{E}[X^1]\|_2^2\right] \\
&\leq \mathbb{E}\left[\|\Pi_M(X^{1/2}) - \mathbb{E}[X^{1/2}]\|_2^2\right] \\
&\leq \mathbb{E}\left[\|X^{1/2} - \mathbb{E}[X^{1/2}]\|_2^2\right] = \sum_k \mathbb{V}\mathrm{ar}[X^{1/2}].
\end{aligned}$$

Hence, it suffices to prove the bound for $X^{1/2}$—i.e., for the Elo update without capping.

Use subscript $\{i, j\}$ to indicate that this pair is chosen in the first game, as before. Let $\{I, J\}$ be a pair of (distinct) indices drawn according to $\boldsymbol{q}$. Then, by the law of total variance,

$$\sum_k \mathbb{V}\mathrm{ar}[X_k^{1/2}] = \sum_k \mathbb{E}[\mathbb{V}\mathrm{ar}_{\{I,J\}}[X_k^{1/2}]] + \sum_k \mathbb{V}\mathrm{ar}[\mathbb{E}_{\{I,J\}}[X_k^{1/2}]].$$

We studied the first term (aka the "unexplained variance") first. First,

$$\sum_k \mathbb{E}[\mathbb{V}\mathrm{ar}_{\{I,J\}}[X_k^{1/2}]] = \sum_k \sum_{i,j} q_{\{i,j\}} \mathbb{V}\mathrm{ar}_{\{i,j\}}[X_k^{1/2}] = \sum_{i,j} q_{\{i,j\}} \sum_k \mathbb{V}\mathrm{ar}_{\{i,j\}}[X_k^{1/2}].$$

We now bound $\mathbb{V}\mathrm{ar}_{\{i,j\}}[X_k^{1/2}]$ over three cases: $k = i$, $k = j$ and $k \notin \{i, j\}$. We have

$$\begin{aligned}
\mathbb{V}\mathrm{ar}_{\{i,j\}}[X_i^{1/2}] &= \mathbb{V}\mathrm{ar}_{\{i,j\}}[X_i^0 - \eta p_{i,j}(X^0)] + \eta^2 \mathbb{V}\mathrm{ar}_{\{i,j\}}\left[\mathrm{Bern}(p_{i,j}(\rho))\right] \\
&\leq \mathbb{V}\mathrm{ar}[X_i^0] - 2\eta \, \mathbb{C}\mathrm{ov}\left[X_i^0, \, \sigma(X_i^0 - X_j^0)\right] + \tfrac{1}{2}\eta^2,
\end{aligned}$$

since the maximal variance of a $[0, 1]$-valued random variable is $\frac{1}{4}$. Analogously,

$$\begin{aligned}
\mathbb{V}\mathrm{ar}_{\{i,j\}}[X_j^{1/2}] &\leq \mathbb{V}\mathrm{ar}[X_j^0] - 2\eta \, \mathbb{C}\mathrm{ov}[X_j^0, \, \sigma(X_j^0 - X_i^0)] + \tfrac{1}{2}\eta^2 \\
&= \mathbb{V}\mathrm{ar}[X_j^0] + 2\eta \, \mathbb{C}\mathrm{ov}\left[X_j^0, \, \sigma(X_i^0 - X_j^0)\right] + \tfrac{1}{2}\eta^2,
\end{aligned}$$

since $\sigma(-z) = 1 - \sigma(z)$. For $k \notin \{i, j\}$,

$$\mathbb{V}\mathrm{ar}_{\{i,j\}}[X_k^{1/2}] = \mathbb{V}\mathrm{ar}[X_k^0].$$

Bringing these three cases together,

$$\sum_k \mathbb{V}\mathrm{ar}_{\{i,j\}}[X_k^{1/2}] \leq \sum_k \mathbb{V}\mathrm{ar}[X_k^0] + \eta^2 - 2\eta \, \mathbb{C}\mathrm{ov}\left[X_i^0 - X_j^0, \, \sigma(X_i^0 - X_j^0)\right].$$

Now, $\sigma : \mathbb{R} \to [0, 1]$ is increasing. So, $\mathbb{C}\mathrm{ov}[Y, \sigma(Y)] \geq 0$ for any random variable $Y$. Moreover,

$$|\sigma(y) - \tfrac{1}{2}| = |\sigma(y) - \sigma(0)| \geq \min z \in [-2M, 2M]|\sigma'(z)| \cdot |y| \geq \tfrac{1}{4}e^{-2M}|y| \quad \text{for all} \quad y \in [-2M, 2M].$$

Since $X_i^0 - X_j^0 \in [-2M, 2M]$, applying this gives

$$\mathbb{C}\mathrm{ov}_{\{i,j\}}\left[X_i^0 - X_j^0, \, \sigma(X_i^0 - X_j^0)\right] \geq \tfrac{1}{4}e^{-2M} \mathbb{V}\mathrm{ar}_{\{i,j\}}[X_i^0 - X_j^0].$$

Plugging this in above,

$$\sum_k \mathbb{V}\mathrm{ar}_{\{i,j\}}[X_k^{1/2}] \leq \sum_k \mathbb{V}\mathrm{ar}[X_k^0] + \eta^2 - \tfrac{1}{2}e^{-2M}\eta \, \mathbb{V}\mathrm{ar}_{\{i,j\}}[X_i^0 - X_j^0].$$

We now average over $\{i, j\}$:

$$\sum_{i,j} q_{\{i,j\}} \sum_k \mathbb{V}\mathrm{ar}_{\{i,j\}}[X_k^{1/2}] \leq \sum_k \mathbb{V}\mathrm{ar}[X_k^0] + \eta^2 - \tfrac{1}{2}e^{-2M}\eta \sum_{i,j} q_{\{i,j\}} \mathbb{V}\mathrm{ar}_{\{i,j\}}[X_i^0 - X_j^0].$$

Variances are (weighted) sums of squares, which leads to a spectral-gap estimate again:

$$\begin{aligned}
\sum_{i,j} q_{\{i,j\}} \mathbb{V}\mathrm{ar}_{\{i,j\}}[X_i^0 - X_j^0] \\
= \sum_{i,j} q_{\{i,j\}} \mathbb{V}\mathrm{ar}_{\{i,j\}}\left[(X_i^0 - \mathbb{E}[X_i^0]) - (X_j^0 - \mathbb{E}[X_j^0])\right] \\
= \sum_{i,j} q_{\{i,j\}} \mathbb{E}\left[\left((X_i^0 - \mathbb{E}[X_i^0]) - (X_j^0 - \mathbb{E}[X_j^0])\right)^2\right] \\
= \mathbb{E}\left[\sum_{i,j} q_{\{i,j\}}\left((X_i^0 - \mathbb{E}[X_i^0]) - (X_j^0 - \mathbb{E}[X_j^0])\right)^2\right] \\
\geq \mathbb{E}\left[\lambda_q \sum_k (X_k^0 - \mathbb{E}[X_k^0])^2\right] = \lambda_q \sum_k \mathbb{V}\mathrm{ar}[X_k^0],
\end{aligned}$$

by applying the Dirichlet characterisation of the spectral gap with $z := X^0 - \mathbb{E}[X^0]$. Thus,

$$\sum_{i,j} q_{\{i,j\}} \sum_k \mathbb{Var}_{\{i,j\}}[X_k^{1/2}] \leq \sum_k \mathbb{Var}[X_k^0] + \eta^2 - \tfrac{1}{2}e^{-2M}\eta\lambda_q \sum_k \mathbb{Var}[X_k^0]$$
$$= (1 - \tfrac{1}{2}e^{-2M}\eta\lambda_q) \sum_k \mathbb{Var}[X_k^0] + \eta^2.$$

We would like to deduce that $\sum_k \mathbb{Var}_\pi[X_k^0] \leq 4e^{2M}\eta/\lambda_q$ now, analogously to before. But, we must remember the second term in the law of total variance (aka the "explained variance"). We have

$$\mathbb{Var}\big[\mathbb{E}_{\{I,J\}}[X_k^{1/2}]\big] \leq \tfrac{1}{2}\eta^2 q_k = \tfrac{1}{2}\eta^2 \sum_\ell q_{\{k,\ell\}}.$$

Indeed, if Player $k$ is picked—i.e., $k \in \{I, J\}$—then $X_k^{1/2}$ moves up/down to one of two values which differ by $\eta$; if Player $k$ is not picked, then $X_k^{1/2}$ does not move. The maximal variance of such a random variable is $\tfrac{1}{2}\eta^2 q_k$, where $q_k = \sum_\ell q_{\{k,\ell\}}$ is the probability that Player $k$ is picked. Hence,

$$\sum_k \mathbb{Var}\big[\mathbb{E}_{\{I,J\}}[X_k^{1/2}]\big] \leq \tfrac{1}{2}\eta^2 \sum_k q_k = \eta^2,$$

noting the double-counting of edges in $\sum_k q_k = \sum_{k,\ell} q_{\{k,\ell\}} = 2$.

Combining the bounds for the unexplained and explained components of the variance,

$$\sum_k \mathbb{Var}[X_k^{1/2}] \leq (1 - \tfrac{1}{2}e^{-2M}\eta\lambda_q) \sum_k \mathbb{Var}[X_k^0] + 2\eta^2.$$

In particular, stationarity implies that $\sum_k \mathbb{Var}_\pi[X_k^{1/2}] = \sum_k \mathbb{Var}_\pi[X_k^0]$, so

$$\sum_k \mathbb{Var}_\pi[X_k^0] = \sum_k \mathbb{Var}_\pi[X_k^1] \leq \sum_k \mathbb{Var}_\pi[X_k^{1/2}] \leq 4e^{2M}\eta/\lambda_q. \qquad \square$$

We controlled the bias, but did not actually argue that it is non-zero.

*Remark* C.2 (Bias in Expected Rating). The uncapped Elo update ($M = \infty$) implies that

$$\mathbb{E}_\pi[X_i] = \mathbb{E}_\pi\big[X_i + \eta \sum_j q_{\{i,j\}}\big(p_{i,j}(\rho) - p_{i,j}(X)\big)\big]$$
$$= \mathbb{E}_\pi[X_i] + \eta\big(\sum_j q_{\{i,j\}} \mathbb{E}_\pi[p_{i,j}(\rho)] - \sum_j q_{\{i,j\}} \mathbb{E}_\pi[p_{i,j}(X)]\big).$$

Therefore,

$$\sum_j q_{\{i,j\}} p_{i,j}(\rho) = \sum_j q_{\{i,j\}} \mathbb{E}_\pi[p_{i,j}(X)].$$

The right-hand side is the expectation of the estimated probability that Player $i$ wins their next game (not conditioning on their opponent) in equilibrium and the left-hand side is the true probability. The equality shows that the win-probability estimator for each player is unbiased.

In the two-player case, this actually implies that $\mathbb{E}_\pi[p_{1,2}(X)] = p_{1,2}(\rho)$. We can deduce from this, however, that the estimated rating is biased, if $\rho \neq (0,0)$. We do this now, but only informally.

The ratings are zero-sum, so it suffices to consider $X_1$ and $\rho_1$. Suppose that $\rho_1 > 0$ and that $\eta$ is small—much smaller than $\rho_1$. Then, the rating $X_1$ concentrates around $\rho_1$. The win-probability function $\sigma$ is strictly convex and increasing in $(0, \infty)$, which suggests that

$$p_{1,2}\big(\mathbb{E}_\pi[X_1]\big) < \mathbb{E}_\pi\big[p_{1,2}(X_1)\big] = p_{1,2}(\rho), \quad \text{and hence} \quad \mathbb{E}_\pi[X_1] \neq \rho.$$

Of course, $\mathbb{P}_\pi[X_1 > 0] \neq 1$. This can be handled using the quantified version of Jensen's inequality and large deviation estimates on $\mathbb{P}_\pi[X_1 < 0]$. If $\rho_1$ is large, then, very roughly, the latter probability is like $e^{-\rho_1^2}$, whilst the quantified difference from equality is like $e^{-\rho_1}$; the latter dominates.

The $n$-player case is more complicated. It is possible that a particular choice of $(\rho, q)$ could lead to certain players' having unbiased estimates. However, they will be biased in general.

The ratings are biased, typically, but the win-probabilities at equilibrium are unbiased in the uncapped setting. Moreover, the real ratings are the only vector $\rho$ giving rise to these win-probabilities.

**Proposition C.3.** *Let* $x, \rho \in \mathbb{R}^n$ *with* $\sum_k \rho_k = 0 = \sum_k x_k$. *Then,*

$$\sum_j q_{\{i,j\}} p_{i,j}(x) = \sum_j q_{\{i,j\}} p_{i,j}(\rho) \quad \text{for all} \quad i \in [k] \quad \text{if and only if} \quad x = \rho.$$

**Proof.** The "if" direction is obvious. We prove the "only if" direction by constructing a minimisation problem over zero-sum vectors in $\mathbb{R}^n$ with the following two properties:

1. it has a unique minimum at $\rho$;

2. $x$ is a minimum if and only if it satisfies the equations in the statement.

Define $f : \mathbb{R}^n \to \mathbb{R}$ by

$$f(x) := \sum_i \left( \tfrac{1}{2} \sum_j q_{\{i,j\}} \log\left(1 + e^{x_i - x_j}\right) - x_i \sum_j q_{\{i,j\}} \left(p_{i,j}(\rho) - \tfrac{1}{2}\right)\right) \quad \text{for} \quad x \in \mathbb{R}^n.$$

Then, $f$ is strictly convex in $\{x \in \mathbb{R}^n \mid \sum_k x_k = 0\}$, and thus has a unique minimiser in that set. We now take the gradient of $f$ and compare it with $0$:

$$\tfrac{\partial f}{\partial x_i}(x) = \tfrac{1}{2} \sum_j q_{\{i,j\}} \left(2 p_{i,j}(x) - 1\right) - \sum_j q_{\{i,j\}} \left(p_{i,j}(\rho) - \tfrac{1}{2}\right) = \sum_j q_{\{i,j\}} p_{i,j}(x) - \sum_j q_{\{i,j\}} p_{i,j}(\rho).$$

Hence, $x$ is a minimum if and only if $\sum_k x_k = 0$ and $\sum_j q_{\{i,j\}} p_{i,j}(x) = \sum_j q_{\{i,j\}} p_{i,j}(\rho)$. $\qquad\square$

Notice that if we only desire estimates on the win-probabilities, then the whole Elo framework is not needed: simply tracking the empirical proportion of games won between each pair of players suffices.

# D   Convergence Proofs: Curvature and Concentration

Our time-averaged MCMC-type concentration estimates rely crucially on *curvature* bounds. We start by introducing curvature, then bounding it in the case of the Elo ratings. We then apply it to obtain concentration results for time-averaged ratings.

## D.1   Curvature Definitions

We introduce the concept of *curvature* for a Markov chain $P$ on a general metric space $(\Omega, d)$.

**Definition D.1** (Transportation Distance)**.** Let $\mu$ and $\pi$ be probability measures on $\Omega$. The *transportation distance* $W_1(\mu, \pi)$ represents the 'best' way to send $\mu$ to $\pi$ so that, on average, points are moved by the smallest distance:

$$W_1(\mu, \pi) := \inf_{\mathbb{Q}} \mathbb{E}_{\mathbb{Q}}\big[d(X, Y)\big],$$

where the infimum is over all couplings $\mathbb{Q}$ of $(\mu, \pi)$—i.e., $X \sim \mu$ and $Y \sim \pi$, marginally, under $\mathbb{Q}$.

**Definition D.2** (Curvature of Markov Chains)**.** The (*Ricci*) *curvature* of $P$ is

$$\kappa_P := \inf_{x, y \in \Omega} \kappa_{x,y} \quad \text{where} \quad \kappa_{x,y} := 1 - W_1(P_{x,\cdot}, P_{y,\cdot})/d(x,y) \quad \text{for} \quad x, y \in \Omega.$$

A standard application of the triangle inequality and iteration establishing the following result.

**Lemma D.3** (Contraction of Distance)**.** *For all measures $\mu$ and $\pi$ and all $t \geq 0$,*

$$W_1(\mu P, \pi P) \leq (1 - \kappa_P) W_1(\mu, \pi) \quad \text{and} \quad W_1(\mu P^t, \pi P^t) \leq (1 - \kappa_P)^t W_1(\mu, \pi).$$

This reduces to the well-known set-up of *path coupling* [12] when $(\Omega, d)$ is a finite graph endowed with the usual graph distance. Our set-up, however, is very different: $(\Omega, d) = (\mathbb{R}^n, \|\cdot\|_2)$.

Exact calculation of the curvature is rarely required. Rather, a particular coupling is analysed, giving an upper bound on the transportation distance and hence, by extension, an upper bound on the curvature. The key is finding as close to optimal a coupling as possible.

The next subsection establishes an upper bound on the curvature of the Elo process. The final subsection of the section develops applies curvature to concentration.

## D.2   Curvature Bounds for the Elo Process

We determine the worst-case rate of contraction rate $1 - \kappa$ starting from ratings $(x, y) \in \Omega^2$: we show that $\kappa \gtrsim \lambda_q$ where $\lambda_q$ is the spectral gap of the auxiliary random walk with rates $(q_{i,j})$.

**Proposition D.4** (Curvature)**.** *Let $\kappa$ denote the curvature of Elo in $\|\cdot\|_2$. Then,*

$$\kappa \geq \tfrac{1}{8} \eta e^{-2M} \lambda_q.$$

*Proof.* Let $x = (x_k)_{k \in [n]} \in \Omega$ and $y = (y_k)_{k \in [n]} \in \Omega$ be two sets of ratings. We want to bound

$$\mathbb{E}_{x,y}[\|X^1 - Y^1\|_2] \leq (1 - \rho)\|x - y\|_2 \quad \text{for some} \quad \rho \geq 0 \quad \text{under some coupling.}$$

First, we observe that we can ignore the projection step in the Elo update, by Proposition B.3:

$$\|X^1 - Y^1\|_2 = \|\Pi_M(X^{1/2}) - \Pi_M(Y^{1/2})\|_2 \leq \|X^{1/2} - Y^{1/2}\|_2.$$

Again, this is using the notation $X^{1/2}$ to indicate the *uncapped* Elo update, and $X^1 = \Pi_M(X^{1/2})$. We thus study $(X^{1/2}, Y^{1/2})$, under the assumption $\|x\|_\infty, \|y\|_\infty \leq M$. We use the *trivial coupling*:

- the same pair $(I, J)$ of players is chosen;
- the result of the match is the same—i.e., the same player wins—in both systems.

This is legitimate because neither the choice of players nor the law of the outcome depends on the current state—the *estimated* probability that $I$ beats $J$ depends on the state, but the *real* probability does not. Write $\mathbb{E}_{\{i,j\}}[\cdot]$ for the law conditional on choosing pair $\{i, j\}$ to play.

Recall that, for $z \in \mathbb{R}^n$ and $i, j \in [n]$, we write $p_{i,j}(z) := \sigma(z_j - z_i)$ for the estimated probability that $i$ beats $j$ using ratings $z$. If Player $i$ beats $j$, then $i$ gains $\eta p_{j,i}$ points; similarly, $i$ loses $\eta p_{j,i}$ points if Player $j$ beats $i$. Observing that $p_{j,i}(x) - p_{j,i}(y) = p_{i,j}(y) - p_{i,j}(x)$, we obtain

$$\mathbb{E}_{\{i,j\}}\big[(X_i^{1/2} - Y_i^{1/2})^2\big] = p_{i,j}(\rho)\big((x_i - y_i) + \eta(p_{j,i}(x) - p_{j,i}(y))\big)^2$$
$$+ (1 - p_{i,j}(\rho))\big((x_i - y_i) - \eta(p_{i,j}(x) - p_{i,j}(y))\big)^2$$
$$= \big((x_i - y_i) - \eta(p_{i,j}(x) - p_{i,j}(y))\big)^2$$
$$= (x_i - y_i)^2 + \eta^2\big(p_{i,j}(x) - p_{i,j}(y)\big)^2 - 2\eta(x_i - y_i)\big(p_{i,j}(x) - p_{i,j}(y)\big).$$

Switching the roles of $i$ and $j$ and using the fact that $p_{i,j} = 1 - p_{j,i}$ again, we obtain

$$\mathbb{E}_{\{i,j\}}\big[(X_j^{1/2} - Y_j^{1/2})^2\big] = (x_j - y_j)^2 + \eta^2\big(p_{j,i}(x) - p_{j,i}(y)\big)^2 - 2\eta(x_j - y_j)\big(p_{j,i}(x) - p_{j,i}(y)\big)$$
$$= (x_j - y_j)^2 + \eta^2\big(p_{j,i}(x) - p_{j,i}(y)\big)^2 + 2\eta(x_j - y_j)\big(p_{i,j}(x) - p_{i,j}(y)\big).$$

For all other $k$—i.e., for $k \notin \{i, j\}$—we have $X_k^{1/2} = x_k$ and $Y_k^{1/2} = y_k$. Hence,

$$\mathbb{E}_{\{i,j\}}\big[\|X^{1/2} - Y^{1/2}\|_2^2\big] - \|x - y\|_2^2$$
$$\leq 2\eta^2\big(p_{i,j}(x) - p_{i,j}(y)\big)^2 - 2\eta\big((x_i - x_j) - (y_i - y_j)\big)\big(p_{i,j}(x) - p_{i,j}(y)\big)$$
$$\leq -\eta\big|(x_i - x_j) - (y_i - y_j)\big|\big|\sigma(x_i - x_j) - \sigma(y_i - y_j)\big|,$$

with the final inequality using the fact that $\sigma$ is 1-Lipschitz and $\eta < \frac{1}{2}$.

We now bound the difference in probabilities. For $\bar{x}, \bar{y} \in \mathbb{R}$ with $\bar{x} \geq \bar{y}$, we have

$$|\sigma(\bar{x}) - \sigma(\bar{y})| \geq \min_{\bar{z} \in [\bar{y},\bar{x}]} |\sigma'(\bar{z})||\bar{x} - \bar{y}| \geq \tfrac{1}{4} e^{-\max\{|\bar{x}|,|\bar{y}|\}}|\bar{x} - \bar{y}|.$$

Combining the previous results,

$$\mathbb{E}_{\{i,j\}}\big[\|X^{1/2} - Y^{1/2}\|_2^2\big] - \|x - y\|_2^2 \leq -\tfrac{1}{4}\eta e^{-\max\{|x_i - x_j|,|y_i - y_j|\}}|(x_i - x_j) - (y_i - y_j)|^2$$
$$\leq -\tfrac{1}{4}\eta e^{-2M}|(x_i - x_j) - (y_i - y_j)|^2 = -\tfrac{1}{4}\eta e^{-2M}|(x_i - y_i) - (x_j - y_j)|^2,$$

using the fact that $\|x\|_\infty, \|y\|_\infty \leq M$. Summing over $(i, j)$, weighted by $q_{i,j}$, gives

$$\mathbb{E}\big[\|X^{1/2} - Y^{1/2}\|_2^2\big] - \|x - y\|_2^2$$
$$= \textstyle\sum_{i,j \in [n]: i < j} q_{\{i,j\}} \mathbb{E}_{\{i,j\}}\big[\|X^{1/2} - Y^{1/2}\|_2^2 - \|x - y\|_2^2\big]$$
$$\leq -\tfrac{1}{4}\eta e^{-2M} \textstyle\sum_{i,j \in [n]: i < j} q_{i,j}|(x_i - y_i) - (x_j - y_j)|^2.$$

Now, applying the Dirichlet characterisation of the spectral gap (Proposition B.1) with $z := x - y$,

$$\mathbb{E}\big[\|X^{1/2} - Y^{1/2}\|_2^2\big] - \|x - y\|_2^2 \leq -\tfrac{1}{4}\eta e^{-2M}\lambda_q\|x - y\|_2^2;$$

note that $\sum_k x_k = 0 = \sum_k y_k$, so $\sum_k z_k = 0$. Jensen's inequality then gives

$$\mathbb{E}\big[\|X^{1/2} - Y^{1/2}\|_2\big] \leq \mathbb{E}\big[\|X^{1/2} - Y^{1/2}\|_2^2\big]^{1/2}$$
$$\leq (1 - \tfrac{1}{4}\eta e^{-2M}\lambda_q)^{1/2}\|x - y\|_2 \leq (1 - \tfrac{1}{8}\eta e^{-2M}\lambda_q)\|x - y\|_2.$$

Hence, recalling that $\|X^1 - Y^1\|_2 \leq \|X^{1/2} - Y^{1/2}\|_2$,

$$\kappa_{x,y} \geq \tfrac{1}{8}\eta e^{-2M}\lambda_q, \quad \text{and so} \quad \kappa = \inf_{x,y \in \Omega} \kappa_{x,y} \geq \tfrac{1}{8}\eta e^{-2M}\lambda_q. \qquad \square$$

### D.3 Concentration Statements

Our goal is to establish concentration of time-averaged statistics of the Elo process. General results of this form as known as "Chernoff-type bounds for Markov chains". There is a great deal of literature on such concentration bounds. The version we state here is most-closely related to Theorem 3.4 and Proposition 3.4 in [35]; see also Theorem 3 in [15], particularly.

**Definition D.5** (Mixing Time). Let $\mu$ and $\pi$ be measures on a state space $\Omega$. Then, the *total-variation (TV)* distance between $\mu$ and $\pi$ is defined to be

$$\|\mu - \pi\|_{\mathrm{TV}} := \sup_A |\mu(A) - \pi(A)|,$$

where the supremum is over measurable subsets of $\Omega$. Let $X = (X_t)_{t \geq 0}$ be a Markov chain on $\Omega$, and let $\pi$ denote its equilibrium distribution. The *(precision-$\varepsilon$) mixing time* is

$$t_{\mathrm{mix}}(\varepsilon) := \inf\{t \geq 0 \mid \max_{x \in \Omega} \|\mathbb{P}_x[X_t \in \cdot] - \pi\|_{\mathrm{TV}} \leq \varepsilon\}.$$

By convention, we abbreviate $t_{\mathrm{mix}} := t_{\mathrm{mix}}(\frac{1}{4})$.

**Theorem D.6** (Concentration). *Let $X = (X_t)_{t \geq 0}$ be a uniformly ergodic, irreducible Markov chain on a state space $\Omega$, started from its equilibrium distribution $\pi$. Let $t_{\mathrm{mix}}$ denote its $\frac{1}{4}$-mixing time.*

*Let $f : \Omega \to \mathbb{R}$ be a bounded function: $\|f\|_\infty < \infty$. Let $\pi(f) := \mathbb{E}_\pi[f] = \int_\Omega f d\pi$ and $\sigma_f^2 := \mathbb{V}\mathrm{ar}_\pi[f] = \int_\Omega (f - \pi(f))^2 d\pi$ denote the mean and variance, respectively, of $f$ under $\pi$.*

*Let $\zeta > 0$ and $t \geq 0$ be an integer. Then,*

$$\mathbb{P}_\pi\big[\big|\tfrac{1}{t}\sum_{s=0}^{t-1} f(X^s) - \pi(f)\big| \geq \zeta\big] \leq 2 \exp\left(-\frac{\zeta^2 t/t_{\mathrm{mix}}}{16(1 + 2t_{\mathrm{mix}}/t)\sigma_f^2 + 80\zeta\|f\|_\infty/t}\right).$$

*In particular, if $t \geq \max\{32 t_{\mathrm{mix}}, \, 40\zeta\sigma_f^2\|f\|_\infty\}$, then*

$$\mathbb{P}_\pi\big[\big|\tfrac{1}{t}\sum_{s=0}^{t-1} f(X^s) - \pi(f)\big| \geq \zeta\big] \leq 2 \exp\big(-\tfrac{1}{20}\sigma_f^{-2} \cdot \zeta^2 t/t_{\mathrm{mix}}\big).$$

*Remark* D.7 (Connection to Curvature). The following description applies to *finite* state spaces $\Omega$; we have to be more careful in our Elo application later, since that state space is uncountably infinite.

It is well known that curvature bounds the spectral gap $\lambda$ and relaxation time $t_{\mathrm{rel}} = 1/\lambda$ in the reversible case: $\lambda \geq \kappa$, and hence $t_{\mathrm{rel}} = 1/\lambda \leq \kappa^{-1}$. However, it also upper-bounds the mixing time without requiring reversibility, with an additional factor depending on the diameter:

$$t_{\mathrm{mix}}(\varepsilon) \leq \kappa^{-1}\big(\log \mathrm{diam}\, \Omega + \log(1/\varepsilon)\big),$$

assuming that $d(x, y) \geq \mathbf{1}\{x \neq y\}$. The argument for this is straight-forward: briefly,

$$\mathbb{P}[X_t \neq Y_t] = \mathbb{E}[\mathbf{1}\{X_t \neq Y_t\}] \leq \mathbb{E}[d(X_t, Y_t)] \leq (1 - \kappa)^t d(X_0, Y_0) \leq e^{-\kappa t}\, \mathrm{diam}\, \Omega,$$

then use the standard TV–coupling relation. This can then be plugged into the previous concentration bound, obtaining exponential decay in $\kappa t/\log \mathrm{diam}\, \Omega$.

Applying the theorem to the Elo process has a number of complications, primarily that it does not have a finite mixing time: given the initial ratings, it is always supported on a certain countable set; thus, its TV distance to equilibrium, which is continuously-supported, is 1 (maximal). We circumnavigate this by introducing a noisy version in the proof. This is detailed later.

**Theorem D.8** (MCMC Convergence of Time Averages). *Let $f : \mathbb{R}^n \to \mathbb{R}$ be a 1-Lipschitz function and $\zeta \in (0, 1)$. Let $\mu_f := \mathbb{E}_\pi[f]$ and $\sigma_f^2 := \mathbb{V}\mathrm{ar}_\pi[f]$ denote, respectively, its mean and variance under $\pi$, the unique invariant distribution of $X$. For $t, T > 0$, denote the time average of $f$ in $[T, T + t - 1]$*

$$A_f^{t,T} := \tfrac{1}{t}\sum_{s=T}^{T+t-1} f(X^s).$$

*Suppose that $\|f\|_\infty \leq \frac{1}{5}t$. Let $C_1, C_2 < \infty$. Then, there exists a constant $C_0 < \infty$, depending only on $C_1, C_2$ and $f$, such that if*

$$\min\{\lambda_q, \, \eta, \, 1/t, \, \zeta\} \geq n^{-C_1} \quad \text{and} \quad \min\{t, T\} \geq C_0 t_\star \quad \text{where} \quad t_\star := e^{2M}\eta^{-1}\lambda_q^{-1} \log n,$$

*then*

$$\mathbb{P}_0\big[\big|\tfrac{1}{t}\sum_{s=T}^{T+t-1} f(X^s) - \mu_f\big| \geq C_0\zeta\big] \leq n^{-C_2} + 2 \exp\big(-\sigma_f^{-2}\zeta^2 t/t_\star\big).$$

*In particular, under these assumptions, taking $\zeta \asymp \sigma_f\sqrt{t_\star \log n/t} = e^M \sigma_f \log n/\sqrt{\eta\lambda_q t}$,*

$$\mathbb{P}\left[\big|A_f^{t,T} - \mu_f\big| \geq C_0 e^M \frac{\sigma_f}{\sqrt{\eta}} \frac{\log n}{\sqrt{\lambda_q t}}\right] \leq n^{-C_2}.$$

*Remark* D.9 (Convergence Rate). If $X^1, X^2, \ldots$ were iid, then we would have $1/\sqrt{t}$ decay. Chernoff bounds for reversible Markov chains require $t$ to be replaced by $\lambda t$, where $\lambda$ is the spectral gap. The idea is that $t_{\mathrm{rel}} = 1/\lambda$ steps of the Markov chain are required to decorrelate terms when near equilibrium. For non-reversible Markov chains, running for $t_\star$ does the job.

Connecting this to the curvature $\kappa$, roughly, $t_\star \lesssim \kappa^{-1} \log \operatorname{diam} \Omega$ if $\kappa > 0$; see Remark D.7. The Elo process is not reversible and we, in essence, bound $t_\star \lesssim \kappa^{-1} \log n$ and $\kappa \asymp \eta \lambda_q$.

### D.4 Concentration Proofs

Remark D.7 connects curvature and the mixing time for *finite* Markov chains: in essence, curvature allows us to bring two copies *exactly* together. The Elo process, which lives in the continuum $\mathbb{R}^n$, does not have this property. We can get them *extremely* close, though. So, morally, the bound of $\kappa^{-1} \log \operatorname{diam} \Omega \asymp \lambda_q^{-1} \log n$ on 'mixing' suggested by curvature feels correct.

We rigorise this idea by adding a small amount of independent noise to the ratings after every step. This noise is then used to couple the two copies once they are extremely close.

*Outline of Proof of Theorem D.8.* The proof has multiple steps, which we outline now.

    **I** Approximate the Elo process by a noisy version and control the error.
    **II** Check that the curvature of the noisy process is at least as good as the original.
    **III** Control the mixing time, and hence concentration, of the noisy process.
    **IV** Use a burn-in to get the process quantitatively close to equilibrium.
    **V** Compare the equilibrium distributions for the original and noisy versions.

Finally, we bring all the piece together to conclude the proof, checking a variety of conditions.   △

We approximate the Elo process by a noisy version. This circumnavigates issues surrounding the discrete support of $X_t$ versus the continuous support of its equilibrium distribution.

All the statements consist of generic constants $C_0, C_1, C_2 < \infty$. Rather than carry all these dependencies, we make the specific (arbitrary) choice $C_1 := 5$ and exhibit a $C_2$ for which this works. The results can easily be extended to the general set-up, albeit with more notation. With this in mind, let $\delta := n^{-24}$.

***Step I: Noisy Version & Error.*** We consider the following noisy version $U$ of the Elo process $X$.

1. Suppose $U^0 = u^0$. Draw $u^{1/3}$ according to an *uncapped* Elo step—i.e., as if $M = \infty$—started from $u^0$. Suppose Players $i$ and $j$ were chosen—i.e., $\{k \in [n] \mid u_k^{1/3} \neq u_k^0\} = \{i, j\}$.

2. Draw $\tilde{u}_i, \tilde{u}_j \sim^{\mathrm{iid}} \operatorname{Unif}([-\sqrt{\delta}, +\sqrt{\delta}])$ independently of all else and set $\tilde{u}_k := 0$ for $k \notin \{i, j\}$. Set
$$u^{2/3} := u^{1/3} + \tilde{u}.$$

3. Define $U^1 := u^1$ by orthogonally projecting $u^{2/3}$ to $[-M, M]^n$ preserving the sum.

Note that this *does not* preserve the zero-sum property, as $\tilde{u}_i \neq \tilde{u}_j$. We see later, though, that it is important—crucial, even— for these additive noise terms to be taken independently.

We must control the error between the noisy and original versions. To do this, we use the trivial coupling, as before: the same players and result is used. The error does not accumulate super-linearly due to the contractive nature of the Elo update step, as we explain now.

Suppose that $(X^0, Y^0) = (x, y)$ and generate $(X^1, Y^1) = (x', y')$ from a single step of the trivial Elo-update coupling. Suppose that Players $i$ and $j$ play. Let $s \in \{0, 1\}$ be the indicator that Player $i$ beats $j$—the 'score'. Then,
$$x_i' = x_i + \eta\big(s - p_{i,j}(x)\big) \quad \text{and} \quad y_i' = y_i + \eta\big(s - p_{i,j}(y)\big).$$
Subtracting the second from the first gives
$$x_i' - y_i' = x_i - y_i - \eta\big(p_{i,j}(x) - p_{i,j}(y)\big).$$

Analogously,
$$x'_j - y'_j = x_i - y_j + \eta\big(p_{i,j}(x) - p_{i,j}(y)\big).$$

Subtracting these,
$$(x'_i - y'_i) - (x'_j - y'_j) = (x_i - y_i) - (x_j - y_j) - 2\eta\big(p_{i,j}(x) - p_{i,j}(y)\big).$$

But, $p_{i,j}(z) = \sigma(z_i - z_j)$ and $\sigma$ is 1-Lipschitz, so
$$x_i - x_j \le y_i - y_j \quad \text{if and only if} \quad p_{i,j}(x) \le p_{i,j}(y)$$

and
$$|p_{i,j}(x) - p_{i,j}(y)| \le |(x_i - x_j) - (y_i - y_j)| = |(x_i - y_i) - (x_j - y_j)|;$$

also, $\eta \le \frac{1}{2}$. Hence,
$$|(x'_i - y'_i) - (x'_j - y'_j)| \le |(x_i - y_i) - (x_j - y_j)| \le |x_i - y_i| + |x_j - y_j|.$$

Also, trivially,
$$|(x'_i - y'_i) + (x'_j - y'_j)| = |(x_i - y_i) + (x_j - y_j)| \le |x_i - y_i| + |x_j - y_j|,$$

since the Elo update is zero-sum. Combining these bounds,
$$|x'_i - y'_i| + |x'_j - y'_j| = \max\big\{|(x'_i - y'_i) - (x'_j - y'_j)|,\ |(x'_i - y'_i) + (x'_j - y'_j)|\big\} \le |x_i - y_i| + |x_j - y_j|.$$

In other words, an Elo update does not increase the $\ell_1$ distance between the two sets of ratings:
$$\|x' - y'\|_1 \le \|x - y\|_1.$$

This actually immediately implies that the Elo process is non-negatively curved in $\ell_1$.

A consequences of this is that that error can only arise from the additive-noise step:
$$\|X^s - U^s\|_1 \le \|X^{s-1/2} - U^{s-1/2}\|_1 \le \|X^{s-1} - U^{s-1}\|_1 + \sqrt{\delta} \le \ldots \le s\sqrt{\delta}.$$

Iterating this completes **Step I**:
$$\frac{1}{t}\sum_{s=0}^{t-1} \|X^s - U^s\|_1 \le \frac{1}{2}t\sqrt{\delta} \le n^{-7} \quad \text{deterministically.} \tag{I}$$

We can now work with the noisy version $U$, rather than the original version $X$. The key statistic for our analysis is the curvature. This is not hurt by adding noise.

***Step II:*** *Curvature.* We use the same coupling as before and the same noise in each process.

(i) Suppose that $(U^0, V^0) = (u^0, v^0)$. Draw $(u^{1/3}, v^{1/3})$ according to a single step of the trivial Elo coupling started from $(u^0, v^0)$. Suppose that Players $i$ and $j$ were chosen.

(ii) Draw a single noise vector $n$—i.e., $\tilde{u}_i, \tilde{u}_j \sim^{\text{iid}} \text{Unif}([-\sqrt{\delta}, +\sqrt{\delta}])$ and $\tilde{u}_k := 0$ for $k \notin \{i, j\}$. Set
$$u^{2/3} := u^{1/3} + \tilde{u} \quad \text{and} \quad v^{2/3} := v^{1/3} + \tilde{u}.$$

(iii) Define $U^1 := u^1$ and $V^1 := v^1$ by projecting to $[-M, M]^n$ preserving the respective sums.

We have already shown that the trivial Elo coupling contracts. The added noise does not hurt:
$$\|u^1 - v^1\|_2 \le \|u^{2/3} - v^{2/3}\|_2 \le \|u^{1/3} - v^{1/3}\|_2.$$

Hence, the two-stage coupling contracts at least as well as the original, completing **Step II**:
$$\mathbb{E}[\|U^1 - V^1\|_2] \le \mathbb{E}[\|U^{1/3} - V^{1/3}\|_2] \le (1 - \kappa)\|U^0 - V^0\|_2. \tag{II}$$

The reason for adding the noise is to be able to couple two systems $U$ and $V$, and hence bound the mixing time. We can then apply the general concentration result of Theorem D.6.

***Step III:*** *Mixing Time.* Suppose that $(U^0, V^0) = (u^0, v^0)$ with $\|u^0 - v^0\|_\infty \le \delta$. From this point, we proceed via a slightly different coupling, replacing (i, ii, iii) with (i', ii', iii'), defined below.

(i′) Suppose that $(U^0, V^0) = (u^0, v^0)$. Draw $(u^{1/3}, v^{1/3})$ according to a single step of the trivial Elo coupling started from $(u^0, v^0)$ *without capping*. Suppose that Players $i$ and $j$ were chosen.

(ii′) Draw $\tilde{u}_k \sim \mathrm{Unif}([-\sqrt{\delta}, +\sqrt{\delta}))$ and set $\tilde{v}_k := \tilde{u}_k + (u_k^{1/3} - v_k^{1/3}) \in [-\sqrt{\delta}, +\sqrt{\delta}] \bmod 2\sqrt{\delta}$ independently for $k \in \{i, j\}$. (Here, "mod $2\sqrt{\delta}$" means "adjusting by additive multiples of $2\sqrt{\delta}$ as necessary so that $\tilde{v}_k \in [-\sqrt{\delta}, +\sqrt{\delta})$".) Set $\tilde{u}_k, \tilde{v}_k := 0$ for $k \notin \{i, j\}$. Set

$$u^{2/3} := u^{1/3} + \tilde{u} \quad \text{and} \quad v^{2/3} := v^{1/3} + \tilde{v}.$$

(iii′) Set $U^1 := u^1 := \Pi_M(u^{2/3})$ and $V^1 := v^1 := \Pi_M(v^{2/3})$.

If $\|u^0 - v^0\|_\infty \le \delta$ and Players $i$ and $j$ play, leading to $(u^{1/3}, v^{1/3})$ under the Elo coupling, then

$$\max\{|u_i^{1/3} - v_i^{1/3}|, \ |u_j^{1/3} - v_j^{1/3}|\} \le 3\delta,$$

regardless of which player won. Hence, in the notation of the coupling,

$$u_k^{2/3} \equiv u_k^{1/3} + \tilde{u}_k = v_k^{1/3} + \tilde{v}_k \equiv v_k^{2/3} \quad \text{for both} \quad k \in \{i, j\} \quad \text{if} \quad \tilde{u}_i, \tilde{u}_j \in [-\sqrt{\delta} + 3\delta, +\sqrt{\delta} - 3\delta].$$

Say that a step is *successful* if this holds and *fails* otherwise. The probability of failure is at most $6\sqrt{\delta}$. If it succeeds, then Players $i$ and $j$ are coupled and the bound of $\delta$ on the $\ell_\infty$ norm is preserved.

We now iterate. Suppose that $\|U^0 - V^0\|_\infty \le \delta$. and Players $i_t$ and $j_t$ are chosen in step $t$. Let

$$\tau_{\mathrm{c}} := \inf\big\{t \ge 0 \ \big| \ \cup_{s \le t}\{i_t, j_t\} = [n]\big\}$$

be the first time that all players have been chosen. If all the first $\tau_{\mathrm{c}}$ steps are successful, then all players' ratings are coupled. Call this event $\mathcal{C}$; then, $\mathbb{P}[\mathcal{C} \mid \tau_{\mathrm{c}} \le t] \ge 1 - 6\sqrt{\delta}t$.

It remains to analyse two times:

- the 'burn-in' time $\tau_{\mathrm{b}}$ to get to $\ell_\infty$ norm at most $\delta$, using coupling (i, ii, iii);
- the remaining 'coupling time' $\tau_{\mathrm{c}}$ started after the burn in, using coupling (i′, ii′, iii′).

The first is easy to handle using curvature:

$$\mathbb{E}[\|U^t - V^t\|_\infty] \le \mathbb{E}[\|U^t - V^t\|_2] \le e^{-\kappa t}\|U^0 - V^0\|_2 \le e^{-\kappa t} \cdot 2Mn.$$

Thus,

$$\mathbb{P}[\tau_{\mathrm{b}} > t] \le \mathbb{P}[\|U^t - V^t\|_\infty > \delta] \le \mathbb{E}[\|U^t - V^t\|_\infty]/\delta \le \delta \quad \text{if} \quad t \ge t_\delta := \kappa^{-1}\log(2Mn/\delta^2).$$

The second is equally easy. Start with $(U^0, V^0)$ such that $|U_i^0 - V_i^0| \ge 1$ for all $i$. Now, if $\|U^t - V^t\|_\infty \le \delta < 1$, then all players must have been chosen. Hence,

$$\mathbb{P}[\tau_{\mathrm{c}} > t \mid \mathcal{C}] \le \mathbb{P}[\|U^t - V^t\|_\infty > \delta] \le \delta \quad \text{if} \quad t \ge t_\delta = \kappa^{-1}\log(2Mn/\delta^2).$$

Let $\tau := \inf\{t \ge 0 \mid U^t = V^t\}$ denote the coalesce time. Putting the above parts together,

$$\mathbb{P}[\tau > 2t_\delta] \le \mathbb{P}[\tau_{\mathrm{b}} > t_\delta] + \mathbb{P}[\tau_{\mathrm{c}} > t_\delta] + \mathbb{P}[\mathcal{C}^c \mid \tau_{\mathrm{c}} \le t_\delta] \le 2\delta + 6\sqrt{\delta}t_\delta.$$

We have $\kappa^{-1} = 8e^{2M}\eta^{-1}\lambda_q^{-1} \le n^{11}$. So, recalling that $\delta = n^{-24}$, we have

$$6\sqrt{\delta}t_\delta \le 6n^{-12} \cdot n^{11} \cdot 72\log n = 432n^{-1}\log n \ll 1.$$

Hence, $2\delta + 6\sqrt{\delta}t_\delta \le \frac{1}{4}$. Also, $\delta = n^{-24}$ and $M \le \frac{1}{2}n$, so

$$t_\delta = \kappa^{-1}\log(2Mn/\delta^2) \le 8e^{2M}\eta^{-1}\lambda_q^{-1} \cdot 50\log n = 400e^{2M}\eta^{-1}\lambda_q^{-1}\log n = 400t_\star.$$

This completes **Step III**:

$$t_\star\big(\tfrac{1}{4}\big) \le t_\star(2\delta + 6\sqrt{\delta}t_\delta) \le 2t_\delta \le 800t_\star = 800e^{2M}\eta^{-1}\lambda_q^{-1}\log n. \tag{III}$$

The general concentration result of Theorem D.6 applies only for a Markov chain started from equilibrium. We start another chain $V$ from equilibrium and use a burn-in to get $U$ and $V$ close, quantified by curvature. The error between the time-averaged $U$- and $V$-sums is then small.

***Step IV: Burn-In.*** Let $V$ be a noisy Elo process, started from its equilibrium distribution—which will differ from $\pi$ slightly. Under the above coupling, which has contraction rate $1 - \kappa$ by **(II)**,

$$\sum_{s=T}^{T+t-1} \mathbb{E}[\|U^s - V^s\|_1] \leq \sum_{s=T}^{T+t-1} \mathbb{E}[\|U^s - V^s\|_2] \leq \sum_{s \geq T} e^{-\kappa s} \cdot 2Mn$$
$$\leq 4Mn\kappa^{-1}e^{-\kappa T} \leq \delta^{1/3} = n^{-8} \quad \text{if} \quad T \geq T_\delta := \kappa^{-1} \log(2Mn\kappa^{-1}/\delta^{1/3}).$$

Analogously to the previous step, this time using also $\kappa^{-1} \leq n^5$, we have

$$T_\delta = \kappa^{-1} \log(2Mn\kappa^{-1}/\delta) \leq 8e^{2M}\eta^{-1}\lambda_q^{-1} \cdot 31 \log n \leq 250e^{2M}\eta^{-1}\lambda_q^{-1} \log n = 250t_\star.$$

Also, $t \geq \kappa^{-1} \geq n$. We can then deduce the estimate desired for **Step IV**:

$$\mathbb{E}\left[\tfrac{1}{t}\sum_{s=T}^{T+t-1}\|U^s - V^s\|_1\right] \leq \delta^{1/3}/t \leq n^{-9} \quad \text{if} \quad T \geq 250t_\star. \tag{IV}$$

***Step V: Equilibrium Distributions.*** The original and noisy Elo processes have different equilibrium distributions; call them $\pi$ and $\tilde{\pi}$, respectively. Let $Y$ and $V$ be an original, respectively noisy, Elo process started from $\pi$, respectively $\tilde{\pi}$. Then, by the strong law of large numbers,

$$|\pi(f) - \tilde{\pi}(f)| = \lim_{k \to \infty} \left|\tfrac{1}{k}\sum_{\ell=L}^{L+k-1}(f(Y^\ell) - f(V^\ell))\right| \quad \text{almost surely} \quad \text{for all} \quad L \in \mathbb{N}.$$

In the curvature proof, we showed a $\ell_2^2$-contraction of $1 - 2\kappa$ for the Elo process; see Proposition D.4. Adding the noise can increase the $\ell_2$ distance squared by at most $10M\sqrt{\delta}$. Hence,

$$d_\ell := \mathbb{E}[\|Y^\ell - V^\ell\|_2^2] \quad \text{satisfies} \quad d_\ell \leq (1 - 2\kappa)d_{\ell-1} + 10M\sqrt{\delta}.$$

Iterating this,

$$d_\ell \leq \cdots \leq 2Mn(1 - 2\kappa)^\ell + 5M\sqrt{\delta}\kappa^{-1} \leq 6M\sqrt{\delta}\kappa^{-1} \quad \text{if} \quad \ell \geq L,$$

for some sufficiently large $L$. By Cauchy–Schwarz, $\mathbb{E}[\|Y^\ell - V^\ell\|_2] \leq \sqrt{d_\ell}$. Hence,

$$\mathbb{E}\left[\tfrac{1}{k}\sum_{\ell=L}^{L+k-1}\|Y^\ell - V^\ell\|_2\right] \leq \tfrac{1}{k}\sum_{\ell=L}^{L+k-1}\sqrt{d_\ell} \leq 3\delta^{1/4}(M/\kappa)^{1/2}.$$

Plugging this in above, for a 1-Lipschitz function $f$, using $\|\cdot\|_1 \leq \sqrt{n}\|\cdot\|_2$, we obtain

$$|\pi(f) - \tilde{\pi}(f)| \leq \delta^{1/4}(10Mn/\kappa)^{1/2}.$$

Finally, we observe that $10Mn/\kappa \leq n^7$, completing **Step V**:

$$|\pi(f) - \tilde{\pi}(f)| \leq n^{7/2}\delta^{1/4} = n^{-5/2}. \tag{V}$$

We bring the previous give steps together to conclude the proof.

*Conclusion of Proof of Theorem D.8.* We use the following notation, inline with the above:

- $X$ is the original Elo process, started from an arbitrary initial condition;
- $U$ is the noisy Elo process, started from the same state as $X$—i.e., $U^0 = X^0$;
- $V$ is the noisy Elo process, started from equilibrium $\tilde{\pi}$.

Recall that $\|f\|_{\mathrm{Lip}} \leq 1$. This with the triangle inequality allows us to control the steps individually:

$$\left|\tfrac{1}{t}\sum_{s=T}^{T+t-1}f(X^s) - \pi(f)\right|$$
$$\leq \tfrac{1}{t}\sum_{s=T}^{T+t-1}\|X^s - U^s\|_1 \qquad\qquad \textbf{Step I}$$
$$+ \tfrac{1}{t}\sum_{s=T}^{T+t-1}\|U^s - V^s\|_1 \qquad\qquad \textbf{Step IV}$$
$$+ \left|\tfrac{1}{t}\sum_{s=T}^{T+t-1}f(V^s) - \tilde{\pi}(f)\right| \qquad \textbf{Step III}$$
$$+ |\tilde{\pi}(f) - \pi(f)| \qquad\qquad\qquad \textbf{Step V}$$

The first term is at most $\tfrac{1}{2}t\sqrt{\delta} \leq n^{-7}$ deterministically by **Step I** and the last is at most $n^{7/2}\delta^{1/4} = n^{-5/2}$ by **Step V**. The second term is at most $\delta^{1/3}/n = n^{-9}$ in expectation by **Step IV**, which we plug into Markov's inequality. We assume that $\zeta \geq n^{-2}$, so that each of the first two terms is smaller

than $\zeta$, and $n^{-5}/\zeta \leq n^{-3}$. The remaining term is controlled via the general concentration result of Theorem D.6, using the bound $t_\star(\frac{1}{4}) \leq 800 t_\star = 800 e^{2M} \eta^{-1} \lambda_q^{-1} \log n$ from **Step III**. Hence,

$$\mathbb{P}\big[\big|\tfrac{1}{t}\sum_{s=T}^{T+t-1} f(X^s) - \pi(f)\big| > 9\zeta\big]$$
$$\leq \mathbb{P}\big[\big\|\tfrac{1}{t}\sum_{s=T}^{T+t-1} U^s - \tfrac{1}{t}\sum_{s=T}^{T+t-1} V^s\big\|_1 > \zeta\big]$$
$$+ \mathbb{P}\big[\big|\tfrac{1}{t}\sum_{s=T}^{T+t-1} f(V^s) - \pi(f)\big| > 5\zeta\big]$$
$$\leq n^{-6} + 2\exp\big(-\sigma_f^{-2}\zeta^2 t/t_\star\big).$$

We can apply Theorem D.6 if $t \geq 25600 t_\star = 32 \cdot 800 t_\star$ due to **Step III** and **Step IV** if $T \geq 250 t_\star$.

Finally, we make a specific choice of $\zeta$. There is already a polynomial term in the error probability, so we want to choose $\zeta$ as small as possible whilst preserving this. Thus, we take

$$\zeta := 2\sigma_f \sqrt{t_\star \log n / t} = 2e^M \log n / \sqrt{\eta \lambda_q t}. \qquad \square$$

### D.5  Deduction of Theorem 2.5 from Theorems 2.7 and D.8

We start by recalling Theorem 2.5 for the reader's convenience, numbered here as Theorem D.10.

**Theorem D.10** (MCMC Estimator of Time-Averaged Ratings; Theorem 2.5). *Denote the time-averaged rating*

$$A_k^{t,T} := \tfrac{1}{t}\sum_{s=T}^{T+t-1} X_k^s \quad \text{for} \quad k \in [n] \quad \text{and} \quad t, T > 0.$$

*Let $C_1, C_2 < \infty$. Then, there exists a constant $C_0 < \infty$, depending only on $C_1$ and $C_2$, such that if*

$$\min\{\lambda_q, \eta, 1/t\} \geq n^{-C_1} \quad \text{and} \quad \min\{t, T\} \geq C_0 t_\star \quad \text{where} \quad t_\star := e^{2M}\eta^{-1}\lambda_q^{-1}\log n,$$

*then*

$$\mathbb{P}\left[\tfrac{1}{n}\|A^{t,T} - \rho\|_2^2 \leq \frac{C_0 e^{4M}}{\lambda_q n}\left(\eta + \frac{(\log n)^2}{\lambda_q t}\right)\right] \geq 1 - n^{-C_2}.$$

*The same result holds for $\tfrac{1}{n}\|A^{t,T} - \rho\|_1$, the average distance in the $\ell_1$ (rather than $\ell_2$) sense.*

*Proof of Theorem 2.5/D.10.* The two terms inside the probability come from the MCMC-convergence error and the equilibrium-distribution bias. We use the abbreviations

$$A_k^t := \tfrac{1}{t}\sum_{s=T}^{T+t-1} X_k^s, \quad \beta_k := |\pi(\Pi_k) - \rho_k| \quad \text{and} \quad \sigma_k^2 := \sigma_{\Pi_k}^2 = \mathbb{V}\mathrm{ar}_\pi[\Pi_k],$$

where $\Pi_k : \mathbb{R}^n \to \mathbb{R}$ is the $k$-th projector. We separate the MCMC and bias parts:

$$\|A^t - \rho\|_2^2 \leq 2\|A^t - \pi(\Pi)\|_2^2 + 2\|\pi(\Pi) - \rho\|_2^2.$$

For the bias part, we simply note for now that

$$\|\pi(\Pi) - \rho\|_2^2 = \sum_k |\pi(\Pi_k) - \rho_k|^2 = \sum_k \beta_k^2 =: n\bar{\beta}^2.$$

We now turn to the MCMC time-averages. Applying Theorem D.8 with $f := \Pi_k$,

$$\mathbb{P}\big[\big|A_k^{t,T} - \pi(\Pi_k)\big| \geq C_0 e^M \sigma_k \log n / \sqrt{\eta \lambda_q t}\big] \leq n^{-C_2},$$

under the assumptions of that theorem. We perform a union bound over the $n$ players:

$$\mathbb{P}\big[\tfrac{1}{n}\|A^{t,T} - \pi(\Pi)\|_2^2 \geq C_0 e^{2M}\bar{\sigma}^2(\log n)^2/(\eta\lambda_q t)\big] \leq n^{-C_2+1} \quad \text{where} \quad \bar{\sigma}^2 := \tfrac{1}{n}\sum_{k\in[n]}\sigma_k^2.$$

These bias and (average) variance terms are exactly what we handled in Theorem 2.7:

$$\bar{\beta}^2 \leq 4e^{4M}\eta/(\lambda_q n) \quad \text{and} \quad \bar{\sigma}^2 \leq 4e^{2M}\eta/(\lambda_q n).$$

The first part of Theorem 2.5 now follows from plugging these in above and some small manipulations.

To obtain the $1/(\lambda_q t)$ decay, we just need to check that $t \gg t_\star$:

$$\frac{t_\star}{t} = \frac{e^{2M}\lambda_q^{-1}\log n}{\eta t} \asymp \frac{e^{2M}\lambda_q^{-1}\log n}{\lambda_q^{-1}(\log n)^2} = \frac{e^{2M}}{\log n} \ll 1. \qquad \square$$

# E   Convergence Proofs: Parallel Matches

Our previous bounds on the bias and the variance included the spectral gap $\lambda_q$ obtained from *sequential* analysis—namely, we used $\sum_e q_e = 1$. These expressions change in the parallel set-up. This has a knock-on effect on the two terms in Theorem 2.5.

We now state an extended version of Theorem 3.3.

**Theorem E.1.** *Denote the time-averaged rating*

$$A_k^{t,T} := \tfrac{1}{t} \sum_{s=T}^{T+t-1} X_k^s \quad for \quad k \in [n] \quad and \quad t, T > 0,$$

*where $X^s$ is the state after $s \geq 0$ Elo rounds. Then, under the conditions of Theorem 2.5,*

$$\mathbb{P}\left[ \tfrac{1}{n}\|A^{t,T} - \rho\|_2^2 \leq \frac{Ce^{4M}}{\lambda_q n/N}\left(\eta + \frac{(\log n)^2}{\lambda_q t}\right)\right] = 1 - o(1),$$

*where $N := \sum_{e \in E} q_e$ is the mean size of the matching; to emphasise, here, $t$ and $T$ count* rounds. *Moreover, there exists a distribution $\tilde{q} = (\tilde{q}_S)_{S \in \mathcal{M}}$ over matchings whose induced distribution $q = (q_e)_{e \in E}$ over pairs satisfies $\lambda_q \geq \tfrac{1}{3}\lambda_{\mathrm{disc}}^\star$. In particular,*

$$\|A^{t,T} - \rho\|_\star \lesssim \frac{e^{2M}\log n}{\sqrt{\lambda_{\mathrm{disc}}^\star n/N}}\frac{1}{\sqrt{\lambda_{\mathrm{disc}}^\star t}} \quad whp \quad if \quad \eta \asymp \frac{(\log n)^2}{\lambda_{\mathrm{disc}}^\star t}.$$

We separate the proof of Theorem E.1 into two parts: the convergence and the existence of a parallelisable $q$ with $\lambda_q \geq \tfrac{1}{3}\lambda_{\mathrm{disc}}^\star$. The "in particular" part follows exactly as for Theorem 2.5.

*Proof of Theorem E.1: Convergence.* Write $\bar{\beta}^2 := \tfrac{1}{n}\|\pi(\Pi) - \rho\|_2^2$ and $\bar{\sigma}^2 := \tfrac{1}{n}\sum_k \mathbb{V}\mathrm{ar}_\pi[\Pi_k]$ for the average $\ell_2$-bias and variance of the ratings, respectively, as we did in the deduction of Theorem 2.5.

The analysis of the bias is unchanged until we average over the choice $\{i, j\}$ of players. Then, we implicitly used $\sum_e q_e = 1$ when averaging $2\eta^2$ over $\{i, j\}$. Letting $N := \sum_e q_e = \mathbb{E}_{S \sim \tilde{q}}[|S|]$ denote the expected size of the random matching $\tilde{q}$ corresponding to $q$, the same analysis gives

$$\mathbb{E}_x\left[\|X^{1/2} - \rho\|_2^2\right] \leq \|x - \rho\|_2^2 + 2N\eta^2 - \tfrac{1}{2}e^{-4M}\eta \sum_{i,j} q_{\{i,j\}}|(x_i - \rho_i) - (x_j - \rho_j)|^2.$$

Inspecting the analysis of the variance to see where $\sum_e q_e = 1$ is used, we get

$$\sum_k \mathbb{V}\mathrm{ar}[X_k^{1/2}] \leq \sum_k \mathbb{V}\mathrm{ar}[X_k^0] + \tfrac{3}{2}N\eta^2 - \tfrac{1}{2}e^{-2M}\eta\lambda_q \sum_k \mathbb{V}\mathrm{ar}[X_k^0].$$

The proofs then proceed as before to give the bounds

$$\bar{\beta}^2 \leq 4e^{4M}\eta N/(\lambda_q n) \quad \text{and} \quad \bar{\sigma}^2 \leq 3e^{2M}\eta N/(\lambda_q n).$$

The argument for the deduction of Theorem 2.5 gives

$$\mathbb{P}\left[\tfrac{1}{n}\|A^{t,T} - \rho\|_2^2 \leq C_0(\bar{\beta}^2 + \bar{\sigma}^2/(\eta\lambda_q t)\right] \geq 1 - n^{-C_2};$$

now, $(t, T)$ in $A^{t,T}$ correspond to the number of rounds and $q$ may have $\sum_e q_e > 1$—and, so, $\lambda_q$ may be larger than $1/n$, but not than $N/n$. Plugging in the new bounds for $\bar{\beta}$ and $\bar{\sigma}$,

$$\mathbb{P}\left[\tfrac{1}{n}\|A^{t,T} - \rho\|_2^2 \leq \frac{C_0 e^{4M}}{\lambda_q n/N}\left(\eta + \frac{(\log n)^2}{\lambda_q t}\right)\right] \geq 1 - n^{-C_2}. \qquad \square$$

*Proof of Theorem E.1: $\lambda_q \gtrsim \lambda_{\mathrm{disc}}^\star$.* Let $q$ be an optimiser of $\lambda_{\mathrm{disc}}^\star$: $\lambda_q = \lambda_{\mathrm{disc}}^\star$. It can be formulated as a semidefinite program [10], so its solution can be approximated arbitrarily well in polynomial time.

Extending $q$ from $E$ to $[n]^2$ by $q_{i,j} := q_{\{i,j\}}\mathbf{1}\{\{i,j\} \in E\}$ defines a symmetric and substochastic matrix with the same Dirichlet form. Adjusting the diagonal has no effect on the Dirichlet form, so we may assume that it is, in fact, stochastic: its row sums are all *exactly* 1. The spectral gp of this $q$ is still $\lambda_q$, by the Dirichlet characterisation (Proposition B.1).

The Birkhoff–von Neumann theorem allows the decomposition of any $n \times n$ doubly stochastic matrix into a convex combination of (at most) $n^2$ permutation matrices in polynomial time:

$$q = \sum_{\ell=1}^{n^2} \alpha_\ell P_{\sigma_\ell}$$

where $\alpha \in [0, 1]^{n^2}$ with $\sum_\ell \alpha_\ell = 1$ and $P_\sigma$ is the permutation matrix for $\sigma \in \mathsf{Symm}(n)$:

$$(P_\sigma)_{i,j} = \mathbf{1}\{j = \sigma(i)\} \quad \text{for all} \quad i, j \in [n].$$

This can be chosen so that each permutation matrix has non-zero entries only over graph edges.

Given a permutation $\sigma_i$, we decompose it into disjoint cycles. These cycles actually correspond to cycles in the graph; we can discard cycles with only one element. Let $v_1, v_2, ..., v_k$ be the vertices of a particular cycle of length $k$. Then, we can decompose the cycle into three matchings:

- one containing all the edges in the cycle of type $\{v_i, v_{i+1}\}$ with odd $i \in \{1, ..., k-1\}$;
- one containing all the edges in the cycle of type $\{v_i, v_{i+1}\}$ with even $i \in \{1, ..., k-1\}$;
- one matching containing the single edge $\{v_k, v_1\}$.

This decomposition gives rise to a procedure to sample matchings in the graph.

1. Sample a permutation $\Sigma$ according to the convex combination: $\mathbb{P}[\Sigma = \sigma_\ell] = \alpha_\ell$ for each $\ell$.

2. For each cycle in $\Sigma$, independently sample one of the three induced matchings listed above, each with probability $1/3$.

The vertices in different cycles in a permutation are distinct. Hence, this procedure does indeed product a matching. Moreover, the probability that a given edge $\{i, j\} \in E$ belongs to the sampled matching is precisely $\frac{1}{3}q_{i,j} = \frac{1}{3}q_{\{i,j\}}$. Hence, we have constructed a matching with associated spectral gap at least $\frac{1}{3}\lambda_q$. But, by assumption, $\lambda_q = \lambda^\star_{\mathrm{disc}}$, completing the proof. $\qquad\square$

