# OpenReview forum: "An Analysis of Elo Rating Systems via Markov Chains"
_NeurIPS.cc/2024/Conference — NeurIPS 2024 poster_

### Official Review · Reviewer_M8kb · 2024-07-11

**Soundness:** 3
**Presentation:** 3
**Contribution:** 3
**Rating:** 7
**Confidence:** 4

**Summary:**

The paper proves results about the Elo rating system, as used for instance in chess, using tools from probability and Markov chain theory. The Bradley-Terry-Luce model assumes a true ranking exists and that the Elo ranking evolves via a Markov chain when two players compete against each other. Although the stationary measure of the Elo process provides a biased estimator of the true ranking, this bias can be sufficiently small to provide a good approximation of the true ranking nonetheless. The main results consist in finite-time l2 bounds between the true ranking and the estimated Elo ranking, that show how the Elo process can approximate the true ranking. Some experimental results are also given.

**Strengths:**

The results proved are of interest, especially as the existing literature on the subject is scarce and in particular there has not been many works studying the Elo process as a Markov chain. The paper is quite well-written and does a good job at conveying the main ideas of the proof. The mathematical content is serious and I have not detected any major mistake in the proofs.

**Weaknesses:**

I do not see any particular weakness to the present manuscript.

**Questions:**

Here are some specific remarks and suggestions:

p.2-3 In Def. 2.1 I think it is bit surprising at first that $\rho$ does not appear explicitely. I suggest precising that $\rho$ is the true ranking and that $I$ beats $J$ with probability $p_{IJ}(\rho)$

p.3 l.3 "$\pi$ the equilibrium distribution": I suggest adding a reference where existence and uniqueness of the invariant distribution is proved

p.3 l.100 the notation is $\succeq$ is not defined. It does not seem to be used elsewhere in the paper.

p.3 l.103 "is order $n$" should be "is \emph{of} order $n$". "This implies $\lambda_q \leq 4/n$ always": could you provide a short argument for this?

p.4 l.146 "Since $n / \lambda_q \geq 1$": I think the authors mean $\lambda_q \leq 4/n$

p.5 "Even an alternative definition of the spectral gap has been proposed [13]": I think this sentence is a bit misleading as it could suggest that the spectral gap of [13], defined as in the manuscript as the second smallest singular value of the Laplacian, can be used to control the convergence to equilibrium of a Markov chain, as is done classically using multiplicative or additive reversibilization. [13] shows however this notion of spectral gap is related to convergence of empirical averages.

p.5 in l.200 and l.202 what is the $\Delta$ sign at the end of the equations?

p.5 l.242 the notation $\tilde{O}$ is not defined

p.15 Prop. B.1 I think there is a missing $1/2$ factor in the Dirichlet characterization of $\lambda_q$.

p.15 l.541 "Eloupdate" should be "Elo update"

p.16 l.548 [29] is a stack exchange post, which to me is not a good reference. I would either give a classical reference or not give any reference at all, given the result stated is simply the convergence theorem on Hilbert spaces

p.16 Appendix C: the notation $X^{1/2}$ is not introduced

p.17 l.589 "We now turn to the bias" should be "We now turn to the variance"

p.18 l.598 I think two factors $1/4$ and $1/2$ have been swapped: the maximal variance of a $[0,1]$ valued random variable is $1/4$, so the $\eta^2$ terms add up to give $\eta^2 / 2$ instead of $\eta^2 / 4$. This error is maintained throughout the proof and gives eventually a factor $4$ instead of $3$ in Theorem 2.7

p.18 l.607: "$z:=x^{0} - \mathbb{E} [X^0]$ should be $z := X^{0} - \mathbb{E} [X^0]$

p.18 l.611-612 " the maximal variance of such a random variable is $q_k /2$: there should be an additional factor $\eta^2$

p.19 l.617 the step-size $\eta$ is missing in this equation

p.21 l.677-679 the notations and remarks given in this paragraph have already been used in Appendix C, so I suggest moving it there

p.21 in the equation after l.679 what is $\pi_{i,j}$?

p.21 l.683 $\bar{z} \in [\bar{y},\bar{x}]$ should be an index of the minimum

p21 in the second equation after l.685 I think the notation $E_{\ast}$ is not defined

p.22 Are ergodicity and irreducibility sufficient for Theorem D.5 to hold? I think the results from [35] require a stronger form of "uniform ergodicity".

p.22 In Theorem D.5 "Let $t_{\mathrm{mix}}$ denote its mixing time. I think it would be worth giving a definition of the mixing time. Also it depends on a parameter $\epsilon$ so the authors should precise if they consider a particular $\epsilon$, e.g. 1/4. Also I find slighlty confusing that the notation $t_{\mathrm{mix}}$ is also used, e.g. in Theorem D.7, to denote an upper bound on the mixing time.

p.22 In Theorem D.7 the result seems to be stated (and is proved) for a $1$-lipschitz function. In general I expect there should be a dependency in the lipschitz constant

p.22 l.717 use only one notation among $\mu_f$ or $\pi(f)$ for the expectation.

p.23 l.738 again the mysterious $\Delta$ sign

p.23 l.741-743 about the generic constants: I haven't checked all the calculations but I think they are correct, however I must say the implicit assumptions made on $\eta, \lambda_q$, etc; are not always super clear. For instance p.25 I do not see why $t \geq \kappa^{-1} \geq n$. ALso why the equation of l/812 establishes a bound in $\delta^{1/3} = n^{-8}$ and then in l.814 it becomes $\delta^{1/6}$?

p.23 l.756 "Let $s \in \{0,1 \}$" there is a slight abuse of notation in that $s$ is also used at the end of the proof as a summation index. Maybe write $S$, although I admit there is little risk of confusing.

p.26 in l.821 there is a missing expectation

p.26 l.829 I think using the lipschitz condition requires applying the triangular inequality first, in which case there should be norms inside the summation for the step I and step IV terms

p.26 l.831 "The second term is at most $\delta^1/3"$: but the step step concluded with abound of $\delta^{1/6} / n = n^{-5}$. This probably does not affect the result, but I think these inequalities based on implicit assumptions changing from line to line are confusing and could be made clearer.

p.26 l.834: the right-hand side of the inequality on $t_{\mathrm{mix}}(1/4)$ is $800 t_{\mathrm{mix}}$ given what precedes, not $t_{\mathrm{mix}} = 800 \ldots$.

p.27 l.844 $\sum_{k} | A^{t,T}_{k} - \rho_j |_1$: write this either as a sum or as an $\ell^1$ norm

p.27 Proof of Theroem 2.5 why introduce the new notation $\Pi_k$?

p.29 l.895 "the single edge $\{ c_k, c_1 \}$": I guess this is in fact $\{ v_k, v_1 \}$

p.29 l.898: I am not sure to understand how the cycles are sampled: is it each with probability $1/3$?

**Limitations:**

The paper addresses the limitations of the results which are ground for future work.

---

> ### Author Rebuttal · Authors · 2024-08-05
>
> We thank the reviewer greatly for their extremely thorough review of our paper. We are pleased they found the results of interest and importance, as well as finding the presentation well-written. No general points were raised, but many (very helpful) minor points were. We just wanted to highlight a few here.
>
> (4) p.3 l.103. This follows from the Dirichlet characterisation (Prop B.1, with the corrected factor of 1/2). Fix $k$ and take $z_i \coloneqq 1\\{i = k\\} - 1/n$. This gives Rayleigh quotient $n/(n-1) \cdot \sum_l q_{k,l}$. Since $\sum_k \sum_l q_{k,l} = 2$, we can choose $k$ so that $\sum_l q_{k,l} \le 2/n$. Hence, $\lambda \le 2/(n-1) \le 4/n$.
>
> (6) p.5. You are correct that this version is for empirical averages, *not* mixing/similar. We shall clarify this.
>
> (7) p.5 in l.200 and l.202. This triangle merely indicates the end of a remark/similar, analogously to a square at the end of a proof. We first saw this used in one of Geoffrey Grimmett's books, and liked it.
>
> (11) p.16 l.548. We weren't aware that this is just the convergence theorem on Hilbert spaces. Functional analysis is far from our area of expertise. And, actually, looking online, we don't see exactly how this relates. If you would be able to provide a reference, that would be very appreciated. \
> We agree a stackexchange post is not the best reference, but we found the proof referenced quite elegant, so we decided to cite it.
>
> (19) p.21. This was supposed to be $p_{i,j}(\rho)$. At one stage, we were using $\pi_{i,j}$ for the true probabilities and $p_{i,j}$ for the estimated ones, but decided it was better to be explicit, writing $p_{i,j}(\cdot)$ where $\cdot$ varies. Apologies for missing this one; it will be corrected.
>
> (22) p.22. You're right: *uniform* ergodicity is required. Fortunately, chains on compact spaces which converge in TV are almost always uniformly ergodic in practice, and this is the case for the noisy Elo chain. In fact, positive curvature immediately implies this. We will address this lack of clarity in the camera-ready version.
>
> (24) p.22 Theorem D.7. You're right again: we meant *1*-Lipschitz functions. This is of course without loss of generality since, if $f$ is L-Lipschitz, then $g(x) := f(x)/L$ is 1-Lipschitz, so this is enough for the proof. But we would need to add in the dependence on the Lipschitz constant  as you rightly point out. We will clarify this in the revised version of our manuscript
>
> (27) p.23 l.741-743 (31) p.26 l.831. Good spot. We changed the definition of $\delta$ (namely, the power of $n$) whilst writing the paper, and have clearly made a couple of mistakes. Hopefully, there is no cause for doubt that the argument works, but we definitely should correct the numbers so that it's all above board.
>
> (32) p.26 l.834. Good spot, we have conflicting definitions of tmix, differing by a factor 800. We shall correct this, making sure that the correct constants appear in the appropriate places.
>
> (36) p.29 l.898. We first sample a permutation $\Sigma$ from the Birkhoff-von Neumann decomposition; we then decompose $\Sigma$ into disjoint cycles; for any disjoint cycle we sample one of the three matchings described in the bullet list in the paper, each with probability $1/3$. So at the end, edges coming from all cycles in $\Sigma$ will appear in the final sampled matching, but not all edges of a given cycle, only a vertex-disjoint subset.

---

> > ### Comment · Reviewer_M8kb · 2024-08-09
> > **Reply to rebutal**
> >
> > I thank the authors for their response and hope my feedback has been helpful.
> >
> > Regarding the "convergence theorem on Hilbert spaces", I actually meant projection, not convergence, and it seems indeed that many textbooks never mention the contraction property of the orthogonal projection, so a reference is not easy to find. Overall this was a very minor point.

---

### Official Review · Reviewer_EfRM · 2024-07-12

**Soundness:** 3
**Presentation:** 3
**Contribution:** 3
**Rating:** 6
**Confidence:** 2

**Summary:**

This paper provides convergence guarantees for the time-averages of a natural dynamics (Elo) modelized after the Bradley-Terry-Luce model of elo. If the system has $n$ players and $t$ is the number of time steps, the error is shown to be $(e^{4M}/\lambda_q)(n^{-\alpha} + \log(n) \sqrt{1/t})$ with probability $1 - n^{-\beta}$ in $\ell_2$-norm, where $\alpha, \beta > 0$ are universal constants (under assumptions on the learning stepsizes), $M > 0$ is the truncation constant and $\lambda_q$ the spectral gap of the matching graph (tournament design).

The authors further investigates the problem of finding efficient matching graphs (tournament design) to accelerate the convergence of the elo process.

Although the analysis is *online* (the elo is adapted along the matches), the convergence guarantees nearly match the best known *offline* guarantees (computing the elo by batches, after all the matches have been played).

**Strengths:**

- The model is clear and well explained.
- The result is pretty difficult. Despite its difficulty, there is a real effort of presentation, to provide intuition and to split the proofs into comprehensible steps (e.g., section D.4).
- The statements are clear and the results are interesting.

**Weaknesses:**

### (minor) Reordering a few results?

I believe that slightly reordering results in section 2.1 would help to streamline the paper.
For instance, remark 2.4 (plus lines 194-195) would be better *after* the statement of Theorem 2.5 in my opinion, because they make more sense once one has read the main statement.

It isn't clear at first glance that Theorems 2.7 and 2.8 are preliminary results of Theorem 2.5, and it made me confused a little bit.
Although they are interesting on their own, it may be worth putting them in a dedicated subsection.

### (minor) Typography

The paper does not exploit the large text width of NeurIPS' template.
For instance:
- at lines 597 and 599, the $\le$ and $=$ are missaligned
- at line 606, the second block of aligned equations do not require a linebreak (from what I can measure). Other equations could enjoy such improvements (e.g., line 829).

**Questions:**

### General questions

- Analysis this Elo system seems so natural to do with a SGD approach. Is there any in the literature?

- Why do the asymptotic convergence guarantees depend on the number of players? Is this an inevitable feature? I would expect that, if the the stepsize is small enough (or vanishing), the system overall converges to the true elo values. Is this not the case?

- From a more SGD viewpoint and looking at the average dynamics (the dynamics in line 18 are a Robbins-Monro scheme of a continuous time gradient descent), is the objective function convex? If so, SGD is known to have a few theoretical guarantees and I am curious to know how it compares with your results.

### Miscellaneous

-  At line 178, you say that Elo doesn't converge in total variation. Is this because the learning stepsize is constant? At the next line, you seem to claim that this comes from the fact that $X^t$ is finitely supported while $\pi$ has continuous support. It seems wrong to me because if $\hat{\mu}_n$ denotes the empirical distribution (finitely supported) obtained by sampling the gaussian $\mu = \mathcal{N}(0, 1)$, I think that
$$
||\hat{\mu}_n - \mu || \rightarrow 0
$$ almost surely, where the norm is the total variation distance.

- You use the terminology "negatively curved" for "contraction" and "non-positively curve" for "non-expansive". Is there a reason for this?

**Limitations:**

- The guarantees only hold under the assumption that *there exists* a true rating of players, $\rho$, itself generating the outcomes of matches. It is known that in general, the strength of human players is not an order, and more rock-paper-scissors like ($i$ beats $j$, $j$ beats $k$ yet $k$ beats $i$).


### Miscellaneous
- Despite the effort provided by the authors to make the proof clear, the review time was too short so that I could provide a technical review of the results, especially because the present work is not exactly my domain specialty. Hence the present review is mostly high level and the confidence level is low.

---

> ### Author Rebuttal · Authors · 2024-08-05
>
> We appreciate the reviewer has found our results interesting. We thank the reviewer for their comments about presentation and typography: we will definitely take them into consideration when preparing the revised version of our manuscript.
>
> We now reply to their general questions.
>
> 1. We have not been able to find an analysis of Elo with an SGD approach in the literature. See also point 3 below.
>
> 2. We are not exactly sure what the reviewer is meaning with this, so we kindly ask them to point out if our reply does not address their question. The convergence rate needs to depend in some way on the number of players, since each player needs to have played at least once for convergence to happen. If the step size vanishes, then yes, the system converges to the true ratings (this is a consequence of Thm 2.5 in our paper). However, a non-vanishing step-size, no matter how small it is, will result in a non-zero bias.
>
> 3. The function is convex and we agree we could have studied Elo using known theoretical guarantees for SGD. However, using off-the-shelf theorems for SGD, as far as we know, results in non-optimal rates and guarantees only in expectation. That does not mean that a more sophisticated analysis from an optimisation point of view could not address these issues, but then we are not sure this would actually yield a simpler analysis. In addition to that, we believe a Markov chain analysis has its own intrinsic value since it allows us to separate the contribution in the convergence rate due to the bias and the mixing time of the underlying process.
>
> Miscellaneous questions:
>
> 1. We do not think the example mentioned by the reviewer is correct. We believe the total variation distance between the empirical distribution of a sequence of Gaussians and the Gaussian cumulative distribution is actually equal to one. For a reference, see the first page of "No Empirical Probability Measure Can Converge in the Total Variation Sense for all Distributions" by Luc Devroye and László Győrfi in The Annals of Statistics, Vol. 18, No. 3 (Sep., 1990), pp. 1496-1499. \
> The reason Elo does not converge in total variation, however, is actually simpler: when we fix the starting point of the process, after t steps, the distribution is supported on a deterministic set of finite size: for each Elo step, you can choose one of the $n \choose 2$ pairs of players and we have at most two outcomes. Assuming the invariant distribution of the process is atomless, then the total variation distance between the distribution of Elo and the invariant distribution cannot converge to zero as t goes to infinity. We further remark that this observation has been made before by Aldous (reference [7] in our paper).
>
> 2. We use the term curvature in the sense of Ollivier because it is a common concept in the analysis of Markov chains. We can try and make our writing more clear for a broader audience.
>
> About the limitation pointed out by the reviewer: we agree the Bradley-Terry-Luce model is not necessarily an accurate representation of reality. It would be very interesting to study extensions of the Bradley-terry-Luce model beyond linear orderings of players, but this is outside the scope of our contribution.

---

> > ### Comment · Reviewer_EfRM · 2024-08-12
> >
> > Thank you.
> > You have addressed most of my concerns (including question 2).
> >
> > For the miscellaneous question 1, I was mistaking convergence in distribution for convergence in TV. For my failed counter-example you can simply set $A_n := \mathbb{R} \setminus \mathrm{supp}(\mu_n)$ and you get $\mu(A_n) = 1 \ne 0 = \mu_n(A_n)$.

---

### Official Review · Reviewer_ZKzd · 2024-07-12

**Soundness:** 3
**Presentation:** 4
**Contribution:** 3
**Rating:** 6
**Confidence:** 2

**Summary:**

The authors analyze Elo ratings under the BTL assumption and show that the Elo update system converges to the true ratings of the player with high probability. They do this analysis for an online setting compared to other work that has analyzed this for the offline setting (where a pool of data is collected). They then show how this leads to a tournament design.

**Strengths:**

Elo is a popular algorithm that is used from chess to sports like football and baseball. As such it is of broad interest and these results show why this method works. The authors do a thorough analysis and provide empirical data to support their conclusions.

**Weaknesses:**

The work analyzes an algorithm that has been popular and so the contribution doesn't bring any new ideas to the improvement of the ratings system. This raises the question about the magnitude of the contribution.

**Questions:**

Due to lack of expertise of the reviewer for this author, this is left blank

**Limitations:**

Yes

---

> ### Author Rebuttal · Authors · 2024-08-05
>
> We thank the reviewer for their comments and in particular for highlighting that our work provides a thorough analysis of an algorithm with broad interest.
>
> About the weakness highlighted, first, we believe that before improving upon a technique, it is important to understand that technique well. Elo is popular, as pointed out, yet has very little theoretical backing. Our paper is a step in this direction, and further work might be inspired by our theoretical understanding to obtain improvements on Elo. In particular, we highlight the biased nature of the Elo updates: it would be nice to design an unbiased update scheme *without* knowledge of the true ratings, but this seems difficult.
>
> Second, our observations about tournament design might be of practical interest to schemes which learn from pairwise comparisons, potentially leading to improvements.

---

> > ### Comment · Reviewer_ZKzd · 2024-08-14
> >
> > Thanks for your response

---

### Official Review · Reviewer_ZrEV · 2024-07-13

**Soundness:** 3
**Presentation:** 2
**Contribution:** 3
**Rating:** 6
**Confidence:** 3

**Summary:**

The paper presents a novel theoretical analysis of the Elo rating system, a well-established method for ranking player skills in online gaming and sports contexts, particularly in chess. The authors analyze the Elo system under the Bradley-Terry-Luce (BTL) model, employing techniques from Markov chain theory to demonstrate that the Elo system can learn model parameters at a competitive rate compared to state-of-the-art methods. The paper also explores the implications of this analysis on the design of efficient tournaments and draws parallels with the fastest-mixing Markov chain problem. The authors provide rigorous proofs and a discussion of limitations, showcasing the robustness of their analytical approach.

**Strengths:**

1. **Theoretical Rigor**: The paper offers a thorough theoretical analysis of the Elo system, contributing to a deeper understanding of its performance under the BTL model. The use of Markov chain theory to analyze convergence rates adds significant value to the existing literature on the Elo ranking system.


2. **Connection to Tournament Design**: The paper's exploration of tournament design and its relation to the fastest-mixing Markov chain problem is a novel contribution, potentially offering practical insights for tournament organizers.

3. **Clarity and Structure**: The paper is well-structured, with clear explanations of the methods used and the results obtained. The inclusion of and detailed proof guidelines and comparisons with related work enhances authors to understand the research.

**Weaknesses:**

1.	The definition of new variables is extensive but less detailed explanation, making this paper is challenging to understand.
2.	The theoretical result in Theorem2.5 is based on a somehow strong assumption of “The (deterministic) time T is a burn-in phase, which allows the Elo ratings to get 'near' the true skills, after which we start averaging”. So, this theorem only claims the convergence rate at the burn-in phase when the Elo ratings may get 'near' the true skills. If the convergence rate from the start phase could be given would be better, even though requires some conditions like the winning rate among players should be enough to make a difference.
3.	Does the convergence rate in this paper definitely outperform those in other papers? In section 2.2, this paper says “Our result improves the constant in front of M”, but the estimation error of this paper is related to (logn)^2 but those of previous works are only related to logn, they are better than this paper’s result.
4.	What’s the technique novelty of this paper? The form of the convergence rate result is very similar to those of previous works.
5.	If the convergence rate is similar/comparable to previous work, the innovation of this paper is limited because there has existed convergence rate analysis of Elo rating system, and the usage of Markov chain theory does not gain significant improvement on convergence rate.

**Questions:**

1.	In definition 2.3, what the dij and deltaij means? There seem lack some detailed explanation of these symbols’ physical meaning.
2.	What the symbol in the “equation” in line 110 mean? This symbol appears frequently in whole paper, but less explanation.

**Limitations:**

Please refer to weaknesses

---

> ### Author Rebuttal · Authors · 2024-08-05
>
> We thank the reviewer for their comments and in particular for highlighting our paper offers "a thorough theoretical analysis of the Elo system". We first address the weaknesses highlighted.
>
> 1. We will try and add further explanations for the various quantities used in the paper in the camera-ready version. Since the camera-ready version allows for an additional page, this should not be too hard.
>
> 2. We believe with some technical work it is possible to obtain a (potential weaker) bound that does not require a burn-in time. This is due to the fact that the burn-in time T is at least a factor of log n smaller than the total "averaging" time t. \
> Perhaps more pertinently, the time T is deterministic, and fully determined by measurable statistics of the system: the spectral gap λ_q of the choices q and the number n of players. It isn't some ill-defined concept of 'near', which can't actually be known.
>
> 3. Points 3-5 essentially ask what are the improvement over previous work. We would like to point out our work is, as far as we know, the first to analyse "online" Elo, which is what is actually used in practice. Previous work obtained similar bounds only for *offline* algorithms for Bradley-Terry-Luce estimation, exploiting ad-hoc algorithms that are not actually commonly used in practice. \
> Using a Markov chain approach allows us to obtain concentration results. Moreover, we believe it allows us a finer understanding of the process. Mixing is perhaps the most important of these, as it measure the time for the system to 'relax' to equilibrium. In other words, after mixing time steps, the system is close to the true ratings whp, regardless of the initial state. Since Elo is a commonly used algorithm in practice, we believe these results are interesting on their own. \
> As a minor point, to reply to point 3, we remark that if M > log log n, then the constant in front of M is actually more important than the additional log(n) factor.
>
> Questions:
>
> 1. $\Delta$ is simply the Laplacian of the Markov chain with transition rates $(q_{i,j})$. This is a standard matrix used in the analysis of Markov chains. You can view $d_{i,j}$ as the "degrees" of the graph with edge weights $(q_{i,j})$.
>
> 2. $x \asymp y$ simply means that there are absolute constants $c_1,c_2$ such that $c_1 x \le y \le c_2 x$. We will add an explanation to the revised version of our paper.

---

> > ### Comment · Reviewer_ZrEV · 2024-08-14
> >
> > Thank you for your detailed responses. This paper focuses on online Elo ratings, whereas previous works have concentrated on offline settings. This problem setting is interesting to me and addresses most of my concerns. Thus, I will maintain my positive score. Additionally, Reference [1], which also analyzes online Elo ratings, may be worth discussing.
> >
> > [1] Yan, Xue, et al. "Learning to identify top elo ratings: A dueling bandits approach." Proceedings of the AAAI Conference on Artificial Intelligence. Vol. 36. No. 8. 2022.

---

> > > ### Author Response · Authors · 2024-08-14
> > >
> > > Thanks to the reviewer for bringing up reference [1].
> > >
> > > We will discuss it in a bit more details in the revised version of our paper, but we wanted to briefly mention that their setup is quite different than our traditional online Elo scenario, and it is rather more similar to active ranking: instead of observing players play against one another according to a fixed schedule decided before any game has been played, in [1] they actively choose the next pair of players to match up based on the outcome of previous matches.

---

> > > > ### Comment · Reviewer_ZrEV · 2024-08-14
> > > >
> > > > Thank you for your explanation! I understand the difference between your work and [1]. However, I am curious about the rationality of the Elo rating system, which is designed to estimate players' skills and dynamically select matches based on Elo scores, in a fixed schedule setting. Using "a fixed schedule decided before any game has been played" seems akin to tournament designs like round-robin or elimination. Under such fixed tournament schedules, the Elo rating system appears unsuitable because the player matching function is not utilized, and other offline methods, such as MLE and directly computing winning rate, could provide more accurate estimations.

---

> > > > > ### Author Response · Authors · 2024-08-14
> > > > >
> > > > > Thanks for your response. We think there is a slight misunderstanding: the primary purpose of Elo is to provide a **live** ranking of players in a sport. Fans and pundits, even players, want this, so they can see how their current performance stacks up against others, and can be used to estimate the chances of future matches. The ratings are updated after a match in a simple, intuitive way. It's used in a variety of sports for exactly this purpose.
> > > > >
> > > > > An offline system is not suited for this: if you recompute the MLE every time a game is played, the changes in the ranking are opaque and not easily understandable. Moreover, if player A plays against B, only the Elo ratings of A and B are affected, while recomputing the MLE will likely result in all players having a different rating.
> > > > >
> > > > > An "active" system, where you have the flexibility to decide who plays who in the next match, is also not well suited in many scenarios (such as in sports analytics) since you often cannot choose who will play next, but at most you can only decide a schedule before you begin observing outcomes of the matches.
> > > > >
> > > > > In any case, what our work shows is that actually Elo still performs very competitively with state-of-the-art offline algorithms, even though it's online and not "active".

---

### Decision · Program_Chairs · 2024-09-25

**Decision:**

Accept (poster)

**Comment:**

The paper studies the Elo rating system, a popular method to calculate relative skills in zero-sum games. The authors assume a BTL model and show that the Elo rating converges fast with high probability.

I have carefully read the paper, the reviews, and the rebuttal. I like this work, and I think the conclusions it provides and the tools it develops are interesting to the community.

Overall, I recommend acceptance.